# DU-CG-STAP Method Based on Sparse Recovery and Unsupervised Learning for Airborne Radar Clutter Suppression

**Bo Zou, Xin Wang, Weike Feng \*, Hangui Zhu and Fuyu Lu**

Air and Missile Defense College, Air Force Engineering University, Xi'an 710051, China;
zoubo1999@126.com (B.Z.); xin_wang_afeu@163.com (X.W.); zhg598@hotmail.com (H.Z.);
lufuyu1017@126.com (F.L.)
**\*** Correspondence: fengweike007@163.com; Tel.: +86-18066883988

**Abstract:** With a small number of training range cells, sparse recovery (SR)-based space–time adaptive processing (STAP) methods can help to suppress clutter and detect targets effectively for airborne radar. However, SR algorithms usually have problems of high computational complexity and parameter-setting difficulties. More importantly, non-ideal factors in practice will lead to the degraded clutter suppression performance of SR-STAP methods. Based on the idea of deep unfolding (DU), a space–time two-dimensional (2D)-decoupled SR network, namely 2DMA-Net, is constructed in this paper to achieve a fast clutter spectrum estimation without complicated parameter tuning. For 2DMA-Net, without using labeled data, a self-supervised training method based on raw radar data is implemented. Then, to filter out the interferences caused by non-ideal factors, a cycle-consistent adversarial network (CycleGAN) is used as the image enhancement process for the clutter spectrum obtained using 2DMA-Net. For CycleGAN, an unsupervised training method based on unpaired data is implemented. Finally, 2DMA-Net and CycleGAN are cascaded to achieve a fast and accurate estimation of the clutter spectrum, resulting in the DU-CG-STAP method with unsupervised learning, as demonstrated in this paper. The simulation results show that, compared to existing typical SR-STAP methods, the proposed method can simultaneously improve clutter suppression performance and reduce computational complexity.

**Keywords:** space–time adaptive processing (STAP); sparse recovery (SR); deep unfolding (DU); cycle-consistent adversarial network (CycleGAN); unsupervised learning

## 1. Introduction

By simultaneously using spatial information and temporal information, the space–time adaptive processing (STAP) method can improve the clutter suppression and moving target detection performance for airborne radar [1,2]. However, to ensure that the loss of the output signal-to-clutter-plus-noise ratio (SCNR) does not exceed 3 dB compared to the optimal case, the number of independent identically distributed (IID) training range cells required by conventional STAP methods is at least twice the system degrees of freedom (DOF) [3]. In practice, non-ideal factors, e.g., a non-uniform ground/sea environment, non-stationary clutter features, complicated platform movements, and array amplitude/phase errors, often make this condition difficult to meet [4–6].

To reduce the requirement of IID training range cells, dimension-reduced STAP methods, rank-reduced STAP methods, direct-data-domain STAP methods, and SR-based STAP methods have been proposed [7–10]. Among these methods, SR-STAP methods can achieve a high-resolution estimation of the clutter spectrum using a small number of IID training range cells. However, most SR algorithms, e.g., the focal under-determined system solver (FOCUSS), alternating direction method of multipliers (ADMM), and fast-converging sparse Bayesian learning (FCSBL) algorithm [11–13] require lots of iterations to obtain the convergent solution, leading to high computational costs, especially when the problem

dimension is high. In addition, in different clutter environments, the appropriate parameter settings are also a difficult problem for SR-STAP methods. Unreasonable parameter settings will affect the convergence speed and accuracy of SR algorithms. More importantly, various non-ideal factors in practical applications will deteriorate the clutter sparsity and make the SR signal model inaccurate, resulting in significant interferences deviating from the clutter ridge in the space–time domain and thus degrading the clutter suppression performance of SR-STAP methods. These problems limit the applications of SR-STAP methods in practice [14–16].

Recently, the deep neural network (DNN)-based deep learning (DL) technique has been developed and applied to various fields. After proper and sufficient training, DNN can obtain a powerful nonlinear transform capacity for many data-processing or feature-mapping problems [17–20]. In addition, after offline training, DNN only needs forward propagation to complete its operations, thus enjoying a high online computing efficiency. These two properties of DNN can help to solve the above-mentioned problems of SR-STAP methods. For example, a STAP method based on convolutional DNN (CNN) is proposed in [21], which uses the nonlinear image enhancement capability of CNN to realize the high-accuracy reconstruction of the clutter spectrum from its low-accuracy counterpart. It is shown in [21] that, compared to some typical SR-STAP methods, the CNN-based STAP method can obtain a higher clutter-suppression performance with lower computational costs.

Unlike classical data-driven-only DNNs, deep unfolding (DU)-based neural networks combine the data-driven method with the model-driven method [22–25]. In DU-Net, a specific iterative algorithm (e.g., an iterative SR algorithm) with given iterations is unfolded into a DNN with the same number of layers, then the parameters involved in this algorithm are optimized by data learning. In other words, the DU-Net is constructed based on the model of an iterative algorithm. Compared to data-driven DNNs, DU-Nets have the advantage of interpretability and compared to model-driven algorithms, DU-Nets have the advantages of convergence speed and accuracy. Hence, DU-Nets also have the capability to solve the problems of SR-STAP methods. For example, the ADMM algorithm is unfolded into a DNN in [26] for the joint estimation of the clutter spectrum and array error parameters. It is shown in [26] that compared to some typical SR-STAP methods, the DU-Net-based STAP method can improve the clutter suppression performance and reduce the computing complexity.

However, although showing promising potential, DNN-based and DU-Net-based STAP methods have some essential problems that need to be solved. For the STAP methods using the nonlinear image enhancement capability of DNNs, the clutter spectrum estimation performance largely depends on the quality of the input data [27], which cannot be guaranteed using conventional spectrum estimation methods, e.g., the Fourier transform method used in [26]. For the DU-Net-based STAP methods that use the SR algorithm and the DNN method jointly, the performance will be seriously degraded when the clutter sparsity is damaged by the non-ideal practical factors. In addition, for both the DNN-based and DU-Net-based STAP methods, it is usually difficult to construct sufficient and complete input-label paired datasets for supervised training in an unknown environment.

To solve these problems, a DU-CG-STAP method with unsupervised learning is proposed in this paper, which cascades a DU-Net and a DNN. The DU-Net, named as 2DMA-Net, is used to process the raw radar data to estimate the clutter spectrum. It is constructed by unfolding a space–time 2D-decoupled SR algorithm with the multiple-measurement vector (MMV) model, i.e., the 2D-MMV-ADMM algorithm. Similar to [28], the self-supervised learning method based on raw radar data is adopted by 2DMA-Net. The DNN, named a cycle-consistent adversarial network (CycleGAN) [29], is used to process the clutter spectrum obtained by 2DMA-Net to filter out the interferences caused by non-ideal factors. It is acting as a nonlinear image enhancement process with an unsupervised training method based on an input-label unpaired dataset. By using DU-Net and DNN simultaneously, the DU-CG-STAP method can realize a fast and accurate estimation of

the clutter spectrum and thus achieve a high clutter suppression and target detection performance for airborne radar.

To summarize, the main contributions of this paper are as follows:

(1)　To reduce the complexity of solving the SR-STAP model for estimating the clutter spectrum, the MMV-ADMM algorithm is space–time 2D-decoupled. To optimize the iteration parameters of 2D-MMV-ADMM, the 2DMA-Net is constructed. To train 2DMA-Net, the L1 regularization loss function and the mean squared error (MSE) loss function are combined, and thus, with only raw radar data, the self-supervised training method is implemented.

(2)　To solve the performance degradation problem of SR-STAP under non-ideal conditions, the clutter spectrum obtained using 2DMA-Net is processed using CycleGAN. The generator of CycleGAN maps the low-accuracy clutter spectrum into a high-accuracy domain to adaptively extract the clutter features and thus suppress the interferences caused by non-ideal factors. With an unpaired dataset, CycleGAN is trained based on the adversarial criterion and the cycle-consistency criterion.

(3)　To generate an accurate clutter spectrum with low complexity, 2DMA-Net and CycleGAN are cascaded to form the DU-CG-STAP method. With raw radar data and theoretical clutter spectrum as the unpaired dataset, the DU-CG-STAP is trained in an unsupervised way.

The rest of this paper is organized as follows. Section 2 establishes the signal model and briefly introduces the SR-STAP method. Section 3 introduces the processing framework, network structure, dataset construction, and training methods of DU-CG-STAP in detail. Section 4 verifies the performance and advantages of the proposed method via various simulations. Section 5 draws conclusions and discusses future work.

## 2. Signal Model

### 2.1. STAP

As shown in Figure 1, consider an airborne phased-array radar moves at a constant speed $v$ along the $y$-axis at an altitude of $H$. The number of elements in the uniform linear array (ULA) is $M$ and the spacing between adjacent array elements is $d$. The angle between the ULA and the airborne moving direction (i.e., the non-side-looking angle) is $\theta_e$. The radar transmits and receives a total of $N$ pulses in a coherent processing interval (CPI) with a pulse repetition interval of $T_r$.

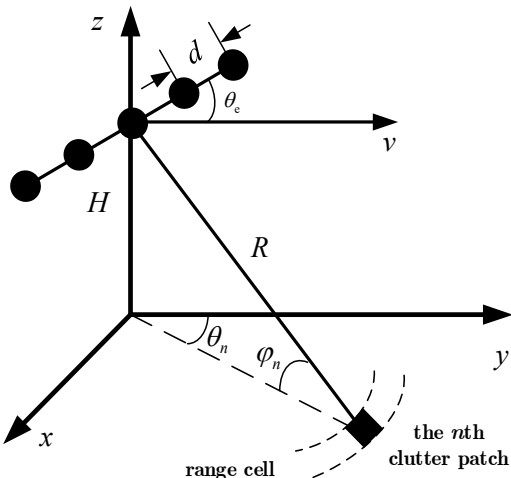

**Figure 1.** Geometry model of airborne radar.

Without considering the effect of range ambiguity, the range ring on the ground/sea surface corresponding to each range cell is supposed to consist of $N_c$ clutter patches with mutually independent scattering coefficients. Thus, the clutter-plus-noise component

contained in the radar-received signal $x$ from the range cell under test (RUT) can be expressed as

$$
\begin{aligned}
x_c + x_n &= \sum_{n=1}^{N_c} \sigma_{c;n} v(f_{c;d,n}, f_{c;s,n}) \odot \alpha(n) + x_n \\
&= \sum_{n=1}^{N_c} \sigma_{c;n} [v_d(f_{c;d,n}) \otimes v_s(f_{c;s,n})] \odot [\alpha_d(n) \otimes \alpha_s(n)] + x_n
\end{aligned}
\tag{1}
$$

where $x_n$ is the noise signal, which can usually be assumed to be complex Gaussian white noise with a mean of 0 and a variance of $\sigma_n^2$, $\otimes$ denotes the Kronecker product, $\sigma_{c;n}$ denotes the scattering coefficient of the $n$th clutter patch, $\alpha_d(n)$ and $\alpha_s(n)$ denotes the weighting vectors related to the temporal and spatial non-ideal factors (e.g., internal clutter motion (ICM), array amplitude error, and array phase error), $v_d(f_{c;d,n})$ and $v_s(f_{c;s,n})$ are the steering vectors of the $n$th clutter patch in the time and space domain, expressed as

$$
\begin{cases}
v_d(f_{c;d,n}) = [1, \exp(j2\pi f_{c;d,n}), \cdots, \exp(j2\pi(N-1)f_{c;d,n})]^T \in \mathbb{C}^{N \times 1} \\
v_s(f_{c;s,n}) = [1, \exp(j2\pi f_{c;s,n}), \cdots, \exp(j2\pi(M-1)f_{c;s,n})]^T \in \mathbb{C}^{M \times 1}
\end{cases}
\tag{2}
$$

where $[\cdot]^T$ denotes transpose operation, $f_{c;d,n}$ and $f_{c;s,n}$ are the Doppler frequency and spatial frequency of the $n$th clutter patch, expressed as

$$
\begin{cases}
f_{c;d,n} = \frac{2vT_r}{\lambda} \cos\theta_n \cos\varphi_n \\
f_{c;s,n} = \frac{d}{\lambda} \cos(\theta_n + \theta_e) \cos\varphi_n
\end{cases}
\tag{3}
$$

where $\varphi_n$ and $\theta_n$ are the elevation and azimuth angles of the $n$th clutter patch, respectively, as shown in Figure 1, and $\lambda$ denotes the signal wavelength.

According to Equation (1), assuming clutter and noise are independent of each other, the clutter-plus-noise covariance matrix (CNCM) can be obtained as

$$
\begin{aligned}
R_I &= R_c + R_n \\
&= E[x_c x_c^H] + E[x_n x_n^H] \\
&= \sum_{n=1}^{N_c} \sigma_{c;n}^2 [v(f_{c;d,n}, f_{c;s,n}) v^H(f_{c;d,n}, f_{c;s,n})] \odot [\alpha(n) \alpha^H(n)] + \sigma_n^2 I_{NM}
\end{aligned}
\tag{4}
$$

where $E[\cdot]$ denotes expectation, $[\cdot]^H$ denotes conjugate transpose, and $I_{NM}$ denotes the unit matrix with a size of $NM \times NM$.

To suppress clutter and detect moving targets, the output of STAP is the inner product of a space–time weighting vector $w$ and the radar-received signal $x$, expressed as

$$
y = w^H x
\tag{5}
$$

To maintain the target power while minimizing the power of clutter and noise, the optimal weighting vector of the space–time filter can be calculated by

$$
w_{opt} = R_I^{-1} v_t / \left[ (v_t)^H R_I^{-1} v_t \right] \in \mathbb{C}^{NM \times 1}
\tag{6}
$$

where $(\cdot)^{-1}$ denotes matrix inverse and $v_t$ is the space–time steering vector of the target.

In practice, the CNCM of the RUT is unknown. In general, a certain number of training range cells that do not include the target are needed to estimate it. To do so, a typical method is to select some range cells near the RUT for training, whereas several range cells adjacent to the RUT on both sides are set as the guard cells to reduce the influence of target contamination [1]. Assuming that the training range cells are IID with the RUT, the CNCM of the RUT can be estimated via the sample matrix inversion (SMI) method [2], expressed as

$$
\hat{R}_I = \frac{1}{L} \sum_{l=1}^{L} x_l x_l^H
\tag{7}
$$

where $L$ is the number of IID training range cells and $x_l$ denotes the radar-received signal from the $l$th training range cell.

According to the RMB criterion [3], the output SCNR loss in dB of the SMI method compared to the optimal STAP method (i.e., the CNCM is known) can be expressed as

$$\text{SCNR}_{\text{loss}} = 10 \times \log\left(\frac{L - O + 2}{L + 1}\right) \tag{8}$$

where $O = MN$ denotes the system DOF.

Equation (8) demonstrates that if the output SCNR loss is required to be less than 3 dB, the number of IID training range cells required by the SMI method is at least twice the system DOF, i.e., $L \geq 2O$, which is difficult to be satisfied in a practical non-uniform and non-stationary clutter environment.

*2.2. SR-STAP*

It can be seen from Equation (1) that, without considering the temporal and spatial non-ideal factors, the clutter signal can be viewed as the superposition of the space–time signals with different spatial and Doppler frequencies. Thus, by discretizing the spatial frequency domain and the Doppler frequency domain into $N_s = \rho_s M$ and $N_d = \rho_d N$ grids with $N_s N_d \gg NM$, the clutter signal can be approximately expressed as

$$x_{\text{c}} = \sum_{i=1}^{N_d} \sum_{j=1}^{N_s} \gamma_{i,j} v\left(f_{\text{d},i}, f_{\text{s},j}\right) = \mathbf{\Phi}\gamma \tag{9}$$

where $f_{\text{d},i}$ is the $i$th ($i = 1, 2, \ldots, N_d$) Doppler frequency, $f_{\text{s},j}$ is the $j$th ($j = 1, 2, \ldots, N_s$) spatial frequency, $v\left(f_{\text{d},i}, f_{\text{s},j}\right)$ is the space–time steering vector corresponding to the $i$-$j$th space–time grid, $\gamma_{i,j}$ denotes the complex amplitude of the $i$-$j$th space–time grid, $\gamma = [\gamma_{1,1}, \gamma_{2,1}, \cdots, \gamma_{N_d,N_s}] \in \mathbb{C}^{N_s N_d \times 1}$ denotes the complex amplitude vector corresponding to all space–time grids, i.e., the space–time amplitude spectrum of the clutter, and $\mathbf{\Phi}$ is a dictionary of space–time steering vectors, expressed as

$$\mathbf{\Phi} = \left[v(f_{\text{d},1}, f_{\text{s},1}), v(f_{\text{d},2}, f_{\text{s},1}), \cdots, v\left(f_{\text{d},N_d}, f_{\text{s},N_s}\right)\right] \in \mathbb{C}^{NM \times N_s N_d} \tag{10}$$

Based on Equation (9), the received signal of the $l$th training range cell without a target can be expressed as

$$x_l = x_{\text{c}}^l + x_{\text{n}}^l = \mathbf{\Phi}\gamma_l + x_{\text{n}}^l \tag{11}$$

Because of the space–time coupling property of clutter, its space–time amplitude spectrum is usually sparse. Hence, the SR-STAP method can estimate the space–time amplitude spectrum of clutter by solving a constrained optimization problem, expressed as

$$\hat{\gamma}_l = \underset{\gamma_l}{\arg\min} \|\gamma_l\|_0, \quad s.t. \|x_l - \mathbf{\Phi}\gamma_l\|_2 \leq \varepsilon \tag{12}$$

where $\|\cdot\|_0$ and $\|\cdot\|_2$ denote the $L_0$ norm and $L_2$ norm of a vector, respectively, and $\varepsilon$ denotes the noise level.

With $L$ training range cells, Equation (12) can be extended to the MMV model [15], expressed as

$$\hat{\mathbf{\Gamma}} = \underset{\mathbf{\Gamma}}{\arg\min} \|\mathbf{\Gamma}\|_{2,0}, \quad s.t. \|\mathbf{X} - \mathbf{\Phi}\mathbf{\Gamma}\|_{\text{F}} \leq \varepsilon \tag{13}$$

where $\mathbf{X} = [x_1, x_2, \ldots, x_L] \in \mathbb{C}^{NM \times L}$, $\mathbf{\Gamma} = [\gamma_1, \gamma_2, \ldots, \gamma_L] \in \mathbb{C}^{N_s N_d \times L}$, $\|\cdot\|_{2,0}$ denotes the $L_0$ norm of the column vector obtained by the $L_2$ norm of each row of a matrix, and $\|\cdot\|_{\text{F}}$ denotes the Frobenius norm of a matrix.

By solving Equation (12) or (13) with a specific SR algorithm, such as FOCUSS, ADMM, or the FCSBL algorithm, the estimation of CNCM can be obtained as

$$\hat{\mathbf{R}}_{\text{I}} = \frac{1}{L}\sum_{l=1}^{L}\sum_{i=1}^{N_d}\sum_{j=1}^{N_s}\left|\gamma_{i,j,l}\right|^2 v\left(f_{\text{d},i}, f_{\text{s},j}\right) v^{\text{H}}\left(f_{\text{d},i}, f_{\text{s},j}\right) + \sigma_{\text{n}}^2 I_{NM} \tag{14}$$

where $\gamma_{l,i,j}$ denotes the complex amplitude of the *i-j*th space–time grid for the *l*th training range cell, i.e., the *i-j-l*th element of $\boldsymbol{\Gamma}$.

Defining $Z_{i,j} = \mathcal{T}(\boldsymbol{\Gamma}_{i,j}) = \sqrt{\frac{1}{L}\sum_{l=1}^{L}\left|\gamma_{i,j,l}\right|^2}$, Equation (14) can be rewritten as

$$\hat{\boldsymbol{R}}_{\mathrm{I}} = \sum_{i=1}^{N_{\mathrm{d}}}\sum_{j=1}^{N_{\mathrm{s}}} Z_{i,j}^2 \boldsymbol{v}(f_{\mathrm{d},i}, f_{\mathrm{s},j})\boldsymbol{v}^{\mathrm{H}}(f_{\mathrm{d},i}, f_{\mathrm{s},j}) + \sigma_{\mathrm{n}}^2 \boldsymbol{I}_{NM} \tag{15}$$

Based on the estimated CNCM, the weighting vector of the space–time filter can be obtained by

$$\hat{w}_{\mathrm{opt}} = \hat{\boldsymbol{R}}_{\mathrm{I}}^{-1}\boldsymbol{v}_{\mathrm{t}} / \left[(\boldsymbol{v}_{\mathrm{t}})^H \hat{\boldsymbol{R}}_{\mathrm{I}}^{-1}\boldsymbol{v}_{\mathrm{t}}\right] \in \mathbb{C}^{NM\times 1} \tag{16}$$

The SR-STAP method can obtain the estimation of CNCM using far fewer IID training range cells than the system DOF, i.e., $L \ll O$. Hence, it has significant advantages over the SMI method in a practical environment.

## 3. DU-CG-STAP

Given the clutter spectrum SR estimation model in Equations (12) or (13), the performance of SR-STAP methods mainly depends on the adopted SR algorithm. Although many effective SR algorithms have been proposed, they have some common problems, e.g., parameter-setting difficulty and high computational complexity. In addition, in practical applications, various non-ideal factors will deteriorate the clutter sparsity and make the SR estimation model inaccurate, resulting in significant interferences deviating from the clutter ridge in the space–time domain and degrading the clutter suppression performance of SR-STAP methods. To solve these problems, a new STAP method, i.e., DU-CG-STAP, is proposed.

The main idea of DU-CG-STAP is to combine an SR-based DU-Net with an image-enhancement DNN. The SR-based DU-Net, namely 2DMA-Net, is used to obtain the clutter spectrum quickly from the raw radar data without parameter tuning. The image-enhancement DNN, namely CycleGAN, is used to process the clutter spectrum obtained by 2DMA-Net to generate an accurate and high-resolution counterpart.

The processing framework of the DU-CG-STAP method is shown in Figure 2. It realizes the nonlinear transform from the raw radar data $\tilde{X}$ to the clutter spectrum $\hat{Z}$, i.e., $\hat{Z} = \mathcal{F}(\tilde{X})$. The key to this method is the DU-CG network, where (1) the 2DMA-Net module is a solving network for the problem in Equation (13) with the network parameter as $\Theta_{\mathrm{A}}$ and the output as the clutter spectrum estimation $\hat{\tilde{\Gamma}} \in \mathbb{C}^{N_{\mathrm{d}}\times N_{\mathrm{s}}\times L}$; (2) the transform module $\mathcal{T}(\cdot)$ completes the single-channel processing of the spectrum $\hat{\tilde{\Gamma}} \in \mathbb{C}^{N_{\mathrm{d}}\times N_{\mathrm{s}}\times L}$ in the range dimension to obtain $\hat{Y} \in \mathbb{C}^{1\times N_{\mathrm{d}}\times N_{\mathrm{s}}}$; (3) the normalization module $\mathcal{N}(\cdot)$ normalizes the clutter spectrum $\hat{Y}$ to obtain $\hat{Y}_{\mathrm{N}} = \mathcal{N}(\hat{Y}) = \hat{Y}/\max(\hat{Y}) \in \mathbb{C}^{1\times N_{\mathrm{d}}\times N_{\mathrm{s}}}$ as the input of $\mathrm{G}_{YZ}$; (4) the generator $\mathrm{G}_{YZ}$ of CycleGAN is the clutter spectrum enhancement network with the parameter as $\Theta_{\mathrm{C}}$ and the output as the normalized clutter spectrum estimation $\hat{Z}_{\mathrm{N}} \in \mathbb{C}^{1\times N_{\mathrm{d}}\times N_{\mathrm{s}}}$; (5) the restoration module $\mathcal{R}(\cdot)$ obtains $\hat{Z} = \mathcal{R}(\hat{Z}_{\mathrm{N}}) = \hat{Z}_{\mathrm{N}}\times\max(\hat{Y}) \in \mathbb{C}^{1\times N_{\mathrm{d}}\times N_{\mathrm{s}}}$, i.e., the final output of the DU-CG network.

To summarize, the procedure of the DU-CG-STAP method is as follows.

Step 1. Implement the offline training of the DU-CG network (including 2DMA-Net and CycleGAN).

Step 2. Input the raw radar data into the trained DU-CG network to obtain the clutter spectrum estimation.

Step 3. Calculate the CNCM and the space–time weighting vector and then conduct clutter suppression and moving target detection.

In the following, the network structure, dataset construction method, and network training method of DU-CG will be introduced in detail.

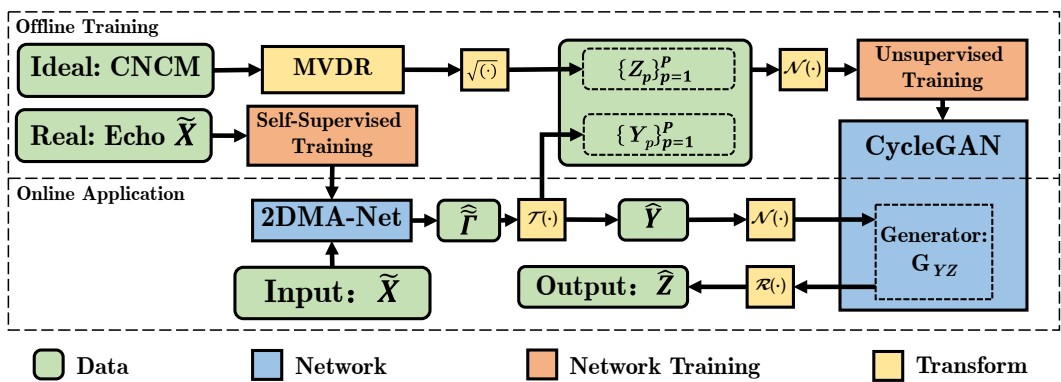

**Figure 2.** Processing framework of DU-CG-STAP.

### 3.1. Network Structure

#### 3.1.1. 2DMA-Net

Since the $L_{2,0}$ norm is a discontinuous function, the complexity of directly solving Equation (13) is quite high. Thus, Equation (13) is usually solved by transforming it into an $L_{2,1}$ convex optimization problem, expressed as

$$\hat{\boldsymbol{\Gamma}} = \arg\min_{\boldsymbol{\Gamma}} \|\boldsymbol{\Gamma}\|_{2,1}, \quad s.t. \|\boldsymbol{X} - \boldsymbol{\Phi}\boldsymbol{\Gamma}\|_F \le \varepsilon \tag{17}$$

Introducing an auxiliary variable $\boldsymbol{R} \in \mathbb{C}^{NM \times L}$, Equation (17) can be transformed into

$$\{\hat{\boldsymbol{\Gamma}}, \hat{\boldsymbol{R}}\} = \arg\min_{\boldsymbol{\Gamma}, \boldsymbol{R}} \left\{ \|\boldsymbol{\Gamma}\|_{2,1} + \frac{1}{2\rho}\|\boldsymbol{R}\|_F^2 \right\} \; s.t. \; \boldsymbol{\Phi}\boldsymbol{\Gamma} + \boldsymbol{R} = \boldsymbol{X} \tag{18}$$

where $\rho > 0$ denotes the regularization factor.

The augmented Lagrange function of Equation (18) is given by

$$\{\hat{\boldsymbol{\Gamma}}, \hat{\boldsymbol{R}}, \hat{\boldsymbol{\Lambda}}\} = \arg\min_{\boldsymbol{\Gamma}, \boldsymbol{R}, \boldsymbol{\Lambda}} \|\boldsymbol{\Gamma}\|_{2,1} + \frac{1}{2\rho}\|\boldsymbol{R}\|_F^2 + \langle \boldsymbol{\Lambda}, \boldsymbol{\Phi}\boldsymbol{\Gamma} + \boldsymbol{R} - \boldsymbol{X}\rangle + \frac{\beta}{2}\|\boldsymbol{\Phi}\boldsymbol{\Gamma} + \boldsymbol{R} - \boldsymbol{X}\|_F^2 \tag{19}$$

where $< \cdot, \cdot >$ denotes the inner product, $\boldsymbol{\Lambda} \in \mathbb{C}^{NM \times L}$ denotes the Lagrange multiplier, and $\beta > 0$ denotes the quadratic penalty factor.

Given an initial value $\left\{\boldsymbol{\Gamma}^{(0)}, \boldsymbol{R}^{(0)}, \boldsymbol{\Lambda}^{(0)}\right\}$, the MMV-ADMM algorithm solves Equation (19) by solving the following three sub-problems alternately with $K$ iterations.

$$\begin{cases} \boldsymbol{R}^{(k)} = \arg\min_{\boldsymbol{R}} \frac{1}{2\rho}\|\boldsymbol{R}\|_F^2 + \frac{\beta}{2}\|\boldsymbol{\Phi}\boldsymbol{\Gamma}^{(k-1)} + \boldsymbol{R} - \boldsymbol{X} + \frac{\boldsymbol{\Lambda}^{(k-1)}}{\beta}\|_F^2 \\ \boldsymbol{\Gamma}^{(k)} = \arg\min_{\boldsymbol{\Gamma}} \|\boldsymbol{\Gamma}\|_{2,1} + \frac{\beta}{2}\|\boldsymbol{\Phi}\boldsymbol{\Gamma} + \boldsymbol{R}^{(k)} - \boldsymbol{X} + \frac{\boldsymbol{\Lambda}^{(k-1)}}{\beta}\|_F^2 \\ \boldsymbol{\Lambda}^{(k)} = \boldsymbol{\Lambda}^{(k-1)} + \beta\left(\boldsymbol{\Phi}\boldsymbol{\Gamma}^{(k)} + \boldsymbol{R}^{(k)} - \boldsymbol{X}\right) \end{cases} \tag{20}$$

where $\boldsymbol{R}^{(k)}$, $\boldsymbol{\Gamma}^{(k)}$, and $\boldsymbol{\Lambda}^{(k)}$ denote the estimation of $\boldsymbol{R}$, $\boldsymbol{\Gamma}$, and $\boldsymbol{\Lambda}$ in the $k$th iteration ($k = 1, 2, \cdots, K$), respectively.

The solutions of Equation (20) can be expressed as [12,30]

$$\begin{cases} \boldsymbol{R}^{(k)} = \frac{\rho\beta}{1+\rho\beta}\left(\boldsymbol{X} - \boldsymbol{\Phi}\boldsymbol{\Gamma}^{(k-1)} - \frac{\boldsymbol{\Lambda}^{(k-1)}}{\beta}\right) \\ \boldsymbol{\Gamma}^{(k)} = \boldsymbol{U}^{(k)} \odot \left(\boldsymbol{\Gamma}^{(k-1)} + \frac{\tau}{\rho\beta}\boldsymbol{\Phi}^{\mathrm{H}}\boldsymbol{R}^{(k)}\right) \\ \boldsymbol{\Lambda}^{(k)} = \boldsymbol{\Lambda}^{(k-1)} + \beta\left(\boldsymbol{\Phi}\boldsymbol{\Gamma}^{(k)} + \boldsymbol{R}^{(k)} - \boldsymbol{X}\right) \end{cases} \tag{21}$$

where $\boldsymbol{U} = u \times 1_{1 \times L} \in \mathbb{C}^{N_d N_s \times L}$, $u = [u_{1,1}, u_{2,1}, \cdots, u_{N_d, N_s}]^T \in \mathbb{C}^{N_d N_s \times 1}$, $u_{i,j} = \frac{\beta\|\boldsymbol{\Gamma}_{i,j}\|_2}{\beta\|\boldsymbol{\Gamma}_{i,j}\|_2 + \tau}$, $\boldsymbol{\Gamma}_{i,j} = [\gamma_{i,j,1}, \gamma_{i,j,2}, \cdots, \gamma_{i,j,L}]$, and $\tau$ is the iteration step size.

It can be seen from Equation (21) that the MMV-ADMM algorithm needs multiple matrix multiplications in each iteration, causing a high computing complexity. To improve the computing speed, the space–time 2D-decoupling process is implemented.

Firstly, the signal matrix $X \in \mathbb{C}^{NM \times L}$, noise matrix $N \in \mathbb{C}^{NM \times L}$, and clutter spectrum matrix $\Gamma \in \mathbb{C}^{N_s N_d \times L}$ are space–time 2D-decoupled and transformed to the three-dimensional (3D) tensor form as $\widetilde{X} \in \mathbb{C}^{N \times M \times L}$, $\widetilde{N} \in \mathbb{C}^{N \times M \times L}$, and $\widetilde{\Gamma} \in \mathbb{C}^{N_d \times N_s \times L}$, respectively. Then, corresponding to the space–time dictionary $\Phi \in \mathbb{C}^{NM \times N_s N_d}$, the spatial dictionary $\Phi_s \in \mathbb{C}^{M \times N_s \times 1}$ and the temporal dictionary $\Phi_d \in \mathbb{C}^{N \times N_d \times 1}$ in the 3D tensor form are constructed. At last, the radar-received signal tensor is expressed as

$$\widetilde{X} = [\![\Phi_d, \widetilde{\Gamma}, \Phi_s^T]\!] + \widetilde{N} \tag{22}$$

where $[\![\cdot]\!]$ denotes the batch multiplication of multiple 3D tensors. For batch multiplication, the matrix slice of each tensor is taken from the third dimension for matrix multiplication. When the third-dimension size of a tensor is one, the batch multiplication takes the same matrix slice each time. For example, the batch multiplication of tensors $a \in \mathbb{C}^{m \times n \times l}$, $b \in \mathbb{C}^{n \times p \times l}$, and $c \in \mathbb{C}^{p \times q \times 1}$ can be simply expressed as $d = [\![a, b, c]\!] \in \mathbb{C}^{m \times q \times l}$.

Based on Equation (22) and the batch multiplication process, the 2D-MMV-ADMM algorithm can be obtained from Equation (21), expressed as

$$\begin{cases} \widetilde{R}^{(k)} = \frac{\rho\beta}{1+\rho\beta}\left(\widetilde{X} - [\![\Phi_d, \widetilde{\Gamma}^{(k-1)}, \Phi_s^T]\!] - \frac{\widetilde{\Lambda}^{(k-1)}}{\beta}\right) \in \mathbb{C}^{N \times M \times L} \\ \widetilde{\Gamma}^{(k)} = \widetilde{U}^{(k)} \odot \left(\widetilde{\Gamma}^{(k-1)} + \frac{\tau}{\rho\beta}[\![\Phi_d^H, \widetilde{R}^{(k)}, \Phi_s^*]\!]\right)_l \in \mathbb{C}^{N_d \times N_s \times L} \\ \widetilde{\Lambda}^{(k)} = \widetilde{\Lambda}^{(k-1)} + \beta\left([\![\Phi_d, \widetilde{\Gamma}^{(k)}, \Phi_s^T]\!] + \widetilde{R}^{(k)} - \widetilde{X}\right) \in \mathbb{C}^{N \times M \times L} \end{cases} \tag{23}$$

where tensors $\widetilde{R}$, $\widetilde{U}$, and $\widetilde{\Lambda}$ are the space–time 2D-decoupled forms of $R$, $U$, and $\Lambda$, respectively.

Given the regularization factor $\rho$, the quadratic penalty factor $\beta$, and the iteration step $\tau$ in advance, the 2D-MMV-ADMM algorithm can obtain the clutter spectrum estimation as $\widetilde{\hat{\Gamma}} = \widetilde{\Gamma}^{(K)}$. Then, the CNCM and the space–time weighting vector can be calculated according to Equations (15) and (16). It can be seen from Equations (21) and (23) that by using the number of multiplications in a single iteration as the indicator, the complexities of the MMV-ADMM algorithm and its space–time 2D-decoupled version are, respectively, $O\left(2NMN_dN_sL + (N_dN_s)^2L + 3NML + N_dN_sL\right)$ and $O(2N_dN_sL + NM(N_d + N_s)L + (N + M)N_dN_sL)$. Compared to the MMV-ADMM algorithm, the 2D-MMV-ADMM algorithm can significantly reduce the computational complexity.

However, in practical applications, the parameter setting for 2D-MMV-ADMM is usually difficult. Unreasonable parameter settings will affect the convergence performance, resulting in high computational complexity and low clutter spectrum estimation accuracy. To solve this problem, based on the idea of DU, the 2D-MMV-ADMM algorithm with $K$ iterations is unfolded into a $K$-layer neural network, i.e., 2DMA-Net, as shown in Figure 3. The data-learning approach is used to obtain the optimal parameters for 2D-MMV-ADMM.

The input, output, and parameters of 2DMA-Net are the signal tensor $\widetilde{X} \in \mathbb{C}^{N \times M \times L}$, the clutter spectrum estimation $\widetilde{\hat{\Gamma}} = \widetilde{\Gamma}^{(K)}$, and $\Theta_A = \left\{\Theta_A^{(k)}\right\}_{k=1}^{K} = \{\rho_k, \beta_k, \tau_k\}_{k=1}^{K}$, respectively. The output of the $k$-th layer of 2DMA-Net is the Lagrange multiplier $\widetilde{\Lambda}^{(k)} \in \mathbb{C}^{N \times M \times L}$, the auxiliary variable $\widetilde{R}^{(k)} \in \mathbb{C}^{N \times M \times L}$, and the clutter spectrum $\widetilde{\Gamma}^{(k)} \in \mathbb{C}^{N_d \times N_s \times L}$. With operations similar to Equation (23), the nonlinear function $F_k\{\cdot\}$ can be expressed as

$$\left\{\widetilde{\Gamma}^{(k)}, \widetilde{R}^{(k)}, \widetilde{\Lambda}^{(k)}\right\} = F_k\left\{\widetilde{X}, \Phi_d, \Phi_s, \widetilde{\Gamma}^{(k-1)}, \widetilde{R}^{(k-1)}, \widetilde{\Lambda}^{(k-1)}, \Theta_A^{(k)}\right\} \tag{24}$$

2DMA-Net is driven by both data training and the theoretical model, hence having the advantages of data adaptability and model interpretability. With optimized network parameters, 2DMA-Net can achieve a higher convergence performance than the 2D-MMV-ADMM

algorithm, thus reducing the computing complexity and improving the performance for estimating the clutter spectrum.

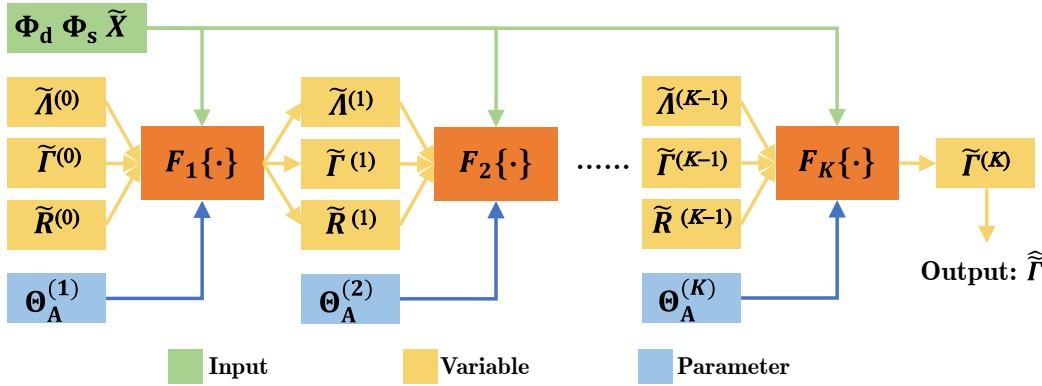

**Figure 3.** Network structure of 2DMA-Net.

3.1.2. CycleGAN

In practice, non-ideal factors will reduce the clutter spectrum estimation accuracy of 2DMA-Net, resulting in some interferences in the space–time domain. To solve this problem, CycleGAN is used as an image-enhancement mapping tool to process the low-accuracy clutter spectrum output of 2DMA-Net. The processing framework of CycleGAN is shown in Figure 4, where the unpaired low-accuracy clutter spectrum $Y_p$ and high-accuracy clutter spectrum $Z_p$ are both the input data. There are two generators of CycleGAN, $G_{YZ}$ and $G_{ZY}$, where $G_{YZ}$ maps the low-accuracy clutter spectrum $Y_p$ into the high-accuracy domain to obtain $\hat{Z}_p$ and $G_{ZY}$ maps the high-accuracy clutter spectrum $Z_p$ into the low-accuracy domain to obtain $\hat{Y}_p$. Discriminators $D_Y$ and $D_Z$ improve the mapping capability of the generators continuously in an adversarial mechanism. After training, the generator $G_{YZ}$ of CycleGAN has the high-accuracy mapping capability for the low-accuracy clutter spectrum.

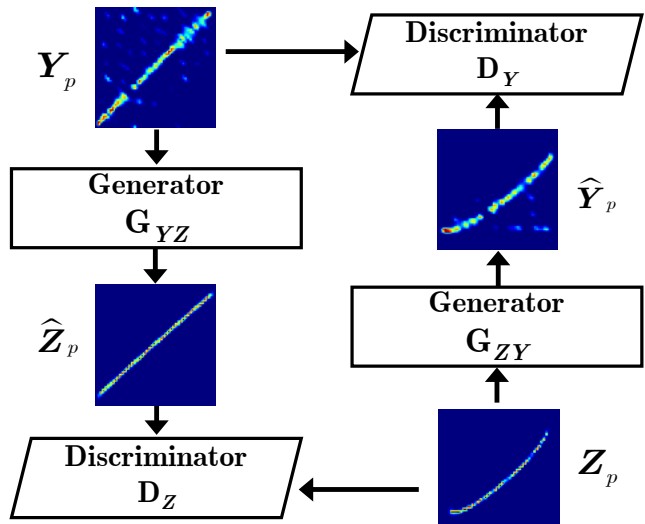

**Figure 4.** Processing framework of CycleGAN.

The network structures of the generator and discriminator of the CycleGAN used in this study are shown in Figure 5, where sigmoid(·) denotes the sigmoid function, Tanh(·) denotes the hyperbolic tangent function, Conv2d denotes the 2D convolution process with the convolution kernel dimension as $c_e \times f_e \times f_e \times n_e$, $c_e$ denotes the number of input channels, $f_e$ denotes the length and width of the convolution kernel, $n_e$ denotes the number of convolution kernels (i.e., the number of output channels), Residual Block denotes the

cascade of two Conv2d layers, and unlike Conv2d that implements the image down-sampling process, ConvTranspose2d implements the image up-sampling process to expand the image size. It should be noted that to better conduct clutter spectrum enhancement tasks and maintain low computational complexity, some appropriate modifications are made to the original network structures of CycleGAN given in [29].

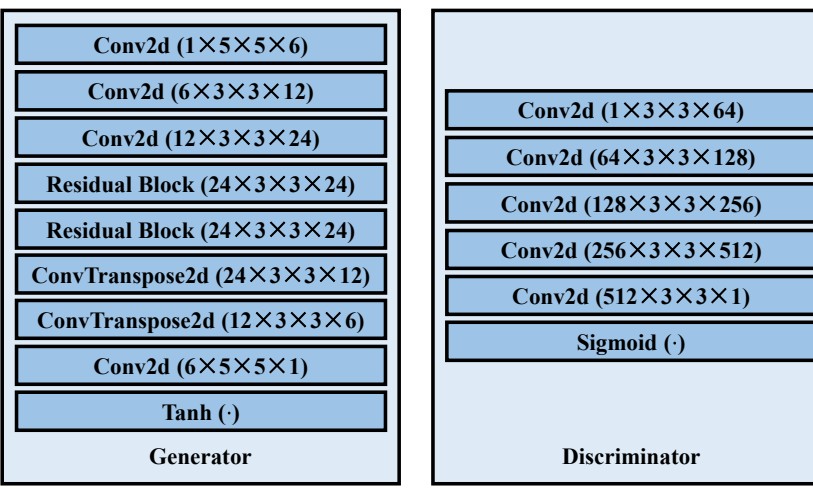

**Figure 5.** Network structures of the generator and discriminator of the CycleGAN in this study.

### 3.2. Dataset Construction

Compared to supervised learning, without using paired training data, unsupervised learning and self-supervised learning can acquire a large number of training data at a low cost. In this study, 2DMA-Net uses a self-supervised learning method and CycleGAN uses an unsupervised learning method, for which the training dataset is constructed with the following three steps.

Step 1. Parameter setting

First, some parameters of the airborne radar system, i.e., signal wavelength $\Lambda$, pulse repetition interval $T_r$, ULA element number $M$, element spacing $d$, CPI pulse number $N$, and the training range cell number $L$, are fixed. In addition, it is set that each range cell consists of $N_c$ clutter patches that are uniformly distributed in the azimuth angle range $[0, \pi]$. The noise power is fixed to $\sigma_n^2 = 1$ and the scattering coefficients of clutter patches obey a complex Gaussian distribution with the amplitude determined by the clutter-to-noise ratio (CNR).

Then, some intermediate parameters are calculated. With $c$ as the speed of light, the maximum unambiguous range is calculated as $R_u = cT_r/2$. Given the space–time frequency range $[f_{s,min}, f_{s,max}]$ and $[f_{d,min}, f_{d,max}]$ and the grid number $N_d = \rho_d N$ and $N_s = \rho_s M$, the spatial dictionary $\mathbf{\Phi}_s \in \mathbb{C}^{M \times N_s \times 1}$ and the temporal dictionary $\mathbf{\Phi}_d \in \mathbb{C}^{N \times N_d \times 1}$ are, respectively, constructed.

Finally, to mimic complicated scenarios, other parameters used to obtain the raw radar data are assumed to be uniformly randomly distributed within specified ranges, i.e., the airplane height $H \in \mathbf{U}[H_{min}, H_{max}]$, the airplane velocity $v \in \mathbf{U}[v_{min}, v_{max}]$, the non-side-looking angle $\theta_e \in \mathbf{U}[\theta_{e,min}, \theta_{e,max}]$, the detection range $R_0 \sim \mathbf{U}[R_{min}, R_u]$, the ICM $\sigma_v \sim \mathbf{U}[\sigma_{v,min}, \sigma_{v,max}]$, the array element amplitude error $\sigma_a \sim \mathbf{U}[\sigma_{a,min}, \sigma_{a,max}]$, the array element phase error $\sigma_p \sim \mathbf{U}[\sigma_{p,min}, \sigma_{p,max}]$, and the CNR CNR$\sim \mathbf{U}($CNR$_{min}$, CNR$_{max})$.

Step 2. Data generating

According to the above settings, $P$ different scenarios with random parameters are simulated and the raw radar data $\tilde{X}_p$ ($p = 1, 2, \ldots, P$) corresponding to each scenario are generated based on Equation (1) and used as the input data for 2DMA-Net. After training 2DMA-Net with the self-supervised method as presented in the following subsection, the

low-accuracy clutter spectrum $Y_p$ is generated corresponding to each set of raw radar data $\widetilde{X}_p$ and used as the input data for CycleGAN.

To train the CycleGAN with the unsupervised method, $P$ different scenarios are simulated with random parameters. Meanwhile, the theoretical CNCM is calculated for each scenario based on Equation (4), where no array amplitude/phase error is contained and thus the spatial weighting vector is fixed as $\boldsymbol{\alpha}_{\mathrm{s}}(n) = \boldsymbol{I}_{M \times 1}$. Then, based on the minimum variance distortionless response (MVDR) algorithm [31], the high-accuracy clutter spectrum $Z_p$ corresponding to each scenario is generated and is also used as the input data for CycleGAN.

Step 3. Dataset partitioning

In Step 2, the generated dataset for 2DMA-Net is $\left\{\widetilde{X}_p\right\}_{p=1}^{P}$ and the generated dataset for CycleGAN is $\left\{Y_p, Z_p\right\}_{p=1}^{P}$. As shown in Figure 6, in this step, the generated datasets are divided into training datasets and validation datasets according to a certain proportion, with the sizes as $P_{\mathrm{train}}$ and $P_{\mathrm{test}}$, respectively.

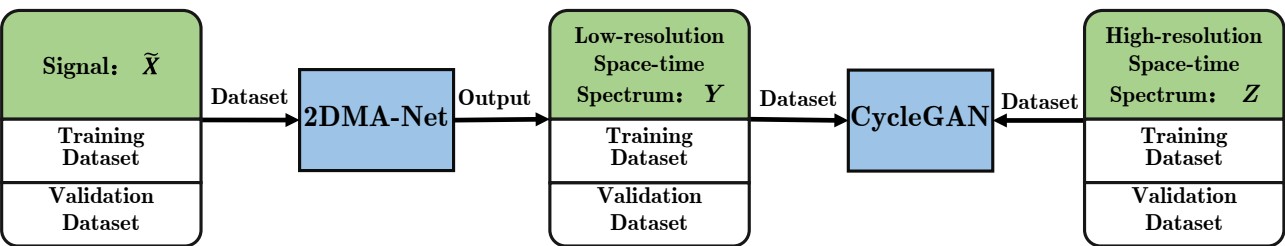

**Figure 6.** Partitioning method of the generated datasets.

*3.3. Network Training*

3.3.1. 2DMA-Net

In most existing DU-Nets, the supervised training method is used, i.e., the output label for each input data is prepared for network training. However, for airborne radar STAP applications, the output label of 2DMA-Net is difficult to obtain as no exact clutter spectrum is available for each set of input raw radar data. A possible solution is to apply the 2D-MMV-ADMM algorithm with fixed manual-tuned parameters and sufficient iterations to solve Equation (22) to obtain the clutter spectrum estimation as the output label for 2DMA-Net. However, with fixed parameters this method cannot guarantee the estimation accuracy for different inputs, resulting in the distortion of output labels. In addition, to obtain convergence, this method needs a lot of iterations, resulting in high computing costs. To solve these problems, the self-supervised training method is adopted by 2DMA-Net without preparing output labels.

With the output clutter spectrum estimation $\hat{\tilde{\boldsymbol{\Gamma}}}_p$ of 2DMA-Net for each set of input raw radar data $\widetilde{X}_p$, the clutter data can be reconstructed as $\hat{\tilde{\boldsymbol{X}}}_p = [\![\boldsymbol{\Phi}_{\mathrm{d}}, \hat{\tilde{\boldsymbol{\Gamma}}}_p, \boldsymbol{\Phi}_{\mathrm{s}}^{\mathrm{T}}]\!]$. Then, the following network loss function is defined for the self-supervised training of 2DMA-Net.

$$\mathcal{L}(\boldsymbol{\Theta}_{\mathrm{A}}) = \frac{1}{P_{\mathrm{train}}} \sum_{p=1}^{P_{\mathrm{train}}} \left( \|\hat{\boldsymbol{R}}_{\mathrm{I},p} - \boldsymbol{R}_{\mathrm{I},p}\|_F^2 + \alpha \|\hat{\tilde{\boldsymbol{\Gamma}}}_p\|_1 \right) \tag{25}$$

where $\boldsymbol{R}_{\mathrm{I},p} = \boldsymbol{X}_p \boldsymbol{X}_p^{\mathrm{H}} / L$, $\hat{\boldsymbol{R}}_{\mathrm{I},p} = \hat{\boldsymbol{X}}_p \hat{\boldsymbol{X}}_p^{\mathrm{H}} / L$, $\boldsymbol{X}_p$ and $\hat{\boldsymbol{X}}_p$ are the matrix forms of the tensors $\widetilde{X}_p$ and $\hat{\tilde{\boldsymbol{X}}}_p$, and $\alpha$ is a constant.

It should be noted that in Equation (25), two functions are combined to define the network loss function of 2DMA-Net. The first function (i.e., MSE loss function) is used to ensure the estimation accuracy of the clutter spectrum with the consideration that the more accurate the estimation of $\hat{\tilde{\boldsymbol{\Gamma}}}_p$, the smaller the difference between $\widetilde{X}_p(\boldsymbol{R}_{\mathrm{I},p})$ and

$\hat{\hat{X}}_p(\hat{R}_{\mathrm{I},p})$. The second function (i.e., $L_1$ regularization loss function) is used to improve the sparsity of the clutter spectrum estimation. If only the MSE loss function is used, the clutter spectrum estimation results may be quite different from the sparse solution of the SR-STAP problem. If only the $L_1$ regularization loss function is used, as the clutter sparsity and the SR estimation model will be seriously damaged by the non-ideal factors (e.g., ICM, element amplitude/phase error, and low CNR), the performance of 2DMA-Net may degrade a lot with significant interferences in the space–time domain. Hence, by using a balancing coefficient $\alpha$, the MSE and $L_1$ regularization combined loss function is used by 2DMV-Net for network training to achieve a high clutter spectrum estimation performance.

Based on the loss function given in Equation (25), with the parameters of each 2DMA-Net layer initialized as $\Theta_{\mathrm{A}} = \{\rho_k = \rho_0, \beta_k = \beta_0, \tau_k = \beta_0\}_{k=1}^K$, the optimal parameters of 2DMA-Net $\Theta_{\mathrm{A}}^* = \{\rho_k^*, \beta_k^*, \tau_k^*\}_{k=1}^K$ can be obtained via the back-propagation method [32,33], expressed as

$$\Theta_{\mathrm{A}}^* = \arg \min_{\Theta_{\mathrm{A}}} \mathcal{L}(\Theta_{\mathrm{A}}) \tag{26}$$

### 3.3.2. CycleGAN

For DNNs using the supervised training method, a paired dataset is required. For airborne radar STAP applications, the practical clutter environment is usually unknown in advance, resulting in difficulties for proper dataset construction. To solve this problem, an unsupervised training method is adopted by CycleGAN. To realize the mutual mapping between the clutter spectra in the low-accuracy domain and the high-accuracy domain via the unpaired dataset, CycleGAN conducts the unsupervised training based on an adversarial criterion and cycle-consistency criterion, which are detailed as follows.

(1)   Adversarial training

Consider that the generator $\mathrm{G}_{YZ}$ can accurately map the low-accuracy clutter spectrum $Y_p$ to the high-accuracy domain to obtain $\hat{Z}_p = \mathrm{G}_{YZ}(Y_p)$ (namely the fake high-accuracy clutter spectrum). Then, it will be difficult for the discriminator $\mathrm{D}_Z$ to distinguish $\hat{Z}_p$ from the true high-accuracy clutter spectrum dataset $\{Z_p\}_{p=1}^P$. The adversarial training process will continuously improve the discriminating capability of $\mathrm{D}_Z$ on the fake and true spectrum and based on the feedback of $\mathrm{D}_Z$, $\mathrm{G}_{YZ}$ will continuously improve its high-accuracy mapping capability on the low-accuracy clutter spectrum. Thus, the following loss function is defined for the generator $\mathrm{G}_{YZ}$ and the discriminator $\mathrm{D}_Z$, expressed as

$$\mathcal{L}_{\mathrm{GAN}}(\mathrm{G}_{YZ}, \mathrm{D}_Z) = E[\log \mathrm{D}_Z(Z_p)] + E[\log(1 - \mathrm{D}_Z(\mathrm{G}_{YZ}(Y_p)))] \tag{27}$$

where $E[\cdot]$ denotes expectation and $\log \mathrm{D}_Z(Z_p)$ and $\log(1 - \mathrm{D}_Z(\mathrm{G}_{YZ}(Y_p)))$ denote the probabilities that the true and fake high-accuracy clutter spectra can be correctly discriminated by $\mathrm{D}_Z$, respectively.

Similarly, the following loss function is defined for the generator $\mathrm{G}_{ZY}$ and the discriminator $\mathrm{D}_Y$, expressed as

$$\mathcal{L}_{\mathrm{GAN}}(\mathrm{G}_{ZY}, \mathrm{D}_Y) = E[\log \mathrm{D}_Y(Y_p)] + E[\log(1 - \mathrm{D}_Y(\mathrm{G}_{ZY}(Z_p)))] \tag{28}$$

where $\log \mathrm{D}_Y(Y_p)$ and $\log(1 - \mathrm{D}_Y(\mathrm{G}_{ZY}(Z_p)))$ denote the probability that the true and fake low-accuracy clutter spectra can be correctly discriminated by $\mathrm{D}_Y$, respectively.

The training process based on the adversarial criterion optimizes the generators $\mathrm{G}_{YZ}/\mathrm{G}_{ZY}$ and the discriminators $\mathrm{D}_Y/\mathrm{D}_Z$ simultaneously, expressed as $\min_{\mathrm{G}_{YZ}} \max_{\mathrm{D}_Z} \mathcal{L}_{\mathrm{GAN}}(\mathrm{G}_{YZ}, \mathrm{D}_Z)$ and $\min_{\mathrm{G}_{ZY}} \max_{\mathrm{D}_Y} \mathcal{L}_{\mathrm{GAN}}(\mathrm{G}_{ZY}, \mathrm{D}_Y)$, i.e., the generators and discriminators will oppositely minimize and maximize the same loss function.

(2)   Cycle-consistency training

The goal of adversarial training is to make it difficult for $\mathrm{D}_Z$ to discriminate $\hat{Z}_p$ from the true high-accuracy clutter spectrum dataset $\{Z_p\}_{p=1}^P$. However, it cannot guarantee

that $Y_p$ and $\hat{Z}_p = \mathrm{G}_{YZ}(Y_p)$ correspond to the same situation. For example, the low-accuracy clutter spectrum in the side-looking case may be transformed by $\mathrm{G}_{YZ}$ into a high-accuracy clutter spectrum in the non-side-looking case. In other words, the adversarial training process only forces $\hat{Z}_p$ to belong to the high-accuracy domain but cannot ensure that $\hat{Z}_p$ is the real desired high-accuracy clutter spectrum counterpart of $Y_p$.

Based on the cycle-consistency criterion, if $Y_p$ can be recovered to the original data by $\mathrm{G}_{YZ}$ and $\mathrm{G}_{ZY}$ successively, i.e., $\mathrm{G}_{ZY}(\mathrm{G}_{YZ}(Y_p)) \approx Y_p$, it can guarantee $\hat{Z}_p$ and $Y_p$ correspond to the same situation. Similarly, for $Z_p$, it has $\mathrm{G}_{YZ}(\mathrm{G}_{ZY}(Z_p)) \approx Z_p$. Hence, the following loss function is defined for the cycle-consistency training, expressed as

$$\mathcal{L}_{\mathrm{cyc}}(\mathrm{G}_{YZ}, \mathrm{G}_{ZY}) = \mathbb{E}\left[\|\mathrm{G}_{ZY}(\mathrm{G}_{YZ}(Y_p)) - Y_p\|_1\right] + \mathbb{E}\left[\|\mathrm{G}_{YZ}(\mathrm{G}_{ZY}(Z_p)) - Z_p\|_1\right] \quad (29)$$

(3)　Full training

To ensure the mapping and correspondence of the clutter spectrum at the same time, the full training process is conducted. Combining the adversarial loss function and the cycle-consistency loss function with their importance balanced by a coefficient $\mu$, the full loss function for CycleGAN is defined as

$$\mathcal{L}(\mathrm{G}_{YZ}, \mathrm{G}_{ZY}, \mathrm{D}_Z, \mathrm{D}_Y, \Theta_\mathrm{C}) = \mathcal{L}_{\mathrm{GAN}}(\mathrm{G}_{ZY}, \mathrm{D}_Y) + \mathcal{L}_{\mathrm{GAN}}(\mathrm{G}_{YZ}, \mathrm{D}_Z) + \mu\mathcal{L}_{\mathrm{cyc}}(\mathrm{G}_{YZ}, \mathrm{G}_{ZY}) \quad (30)$$

Then, by using the Glorot method [34,35] for initialization, the optimal network parameters of CycleGAN can be obtained via the back-propagation method, expressed as

$$\Theta_\mathrm{C}^* = \arg\min_{\mathrm{G}_{YZ}, \mathrm{G}_{ZY}} \max_{\mathrm{D}_Z, \mathrm{D}_Y} \mathcal{L}(\mathrm{G}_{YZ}, \mathrm{G}_{ZY}, \mathrm{D}_Z, \mathrm{D}_Y, \Theta_\mathrm{C}) \quad (31)$$

## 4. Experiment Results

In this section, the performance of the proposed DU-CG-STAP method is verified and compared with three typical SR-STAP methods, i.e., MMV-FOCUSS-STAP, MMV-FCSBL-STAP, and MMV-ADMM-STAP, via various simulations with the parameters shown in Table 1, which are set according to their typical values [13,15,21].

**Table 1.** Simulation parameters.

| Parameter | Notation | Value |
|---|---|---|
| Element spacing | $d$ | 0.1 m |
| Signal wavelength | $\lambda$ | 0.2 m |
| Pulse repetition interval | $T_\mathrm{r}$ | 0.5 ms |
| ULA Element number | $M$ | 10 |
| CPI Pulse number | $N$ | 10 |
| Training range cell number | $L$ | 2 |
| Spatial frequency range | $[f_{\mathrm{s,min}}, f_{\mathrm{s,max}}]$ | $[-0.5, 0.5]$ |
| Doppler frequency range | $[f_{\mathrm{d,min}}, f_{\mathrm{d,max}}]$ | $[-0.5, 0.5]$ |
| Number of spatial frequencies | $N_\mathrm{s}$ | 50 |
| Number of Doppler frequencies | $N_\mathrm{d}$ | 50 |
| Number of clutter patches | $N_\mathrm{c}$ | 181 |
| Airplane height | $H$ | $U[8, 15]$ km |
| Airplane velocity | $v$ | $U[70, 120]$ m/s |
| Detection range | $R_0$ | $U[15\ \mathrm{km}, R_\mathrm{u}]$ |
| Non-side-looking angle | $\theta_\mathrm{e}$ | $U[-30, 30]$ ° |
| Clutter-to-noise-ratio | CNR | $U[30, 50]$ dB |
| Internal clutter motion | $\sigma_\mathrm{v}$ | $U[0, 1]$ m/s |
| Element amplitude error | $\sigma_\mathrm{a}$ | $U[0, 0.2]$ |
| Element phase error | $\sigma_\mathrm{p}$ | $U[0, 10]$ ° |
| Size of training dataset | $P_{\mathrm{train}}$ | 10,000 |
| Size of validation dataset | $P_{\mathrm{test}}$ | 2000 |

In MMV-FOCUSS-STAP, the number of iterations is set as 200 and the sparsity parameter is set as 0.2. In MMV-FCSBL-STAP, the number of iterations is set as 30 and the noise

variance is initialized as $10^{-5}$. In MMV-FCSBL-STAP, the parameters are set as $\rho = 0.5$, $\beta = 0.2$, $\tau = 0.04$, and $K = 2000$. In the self-supervised training of 2DMA-Net, the coefficient of the $L_1$ regularization loss function, the number of network layers, the initial learning rate, and the training epoch are set as $\alpha = 0.01$, $K = 30$, $10^{-4}$, and 500, and the parameters of each layer are initialized as $\Theta_A = \{\rho_k = 0.5, \beta_k = 0.2, \tau_k = 0.04\}_{k=1}^{30}$. In the unsupervised training of CycleGAN, the coefficient of the cycle-consistency loss, the initial learning rate, and the training epoch are set as $\mu = 10$, $2 \times 10^{-5}$, and 500, respectively.

*4.1. Network Convergence Analysis*

In this subsection, the convergence results of network training are presented. Figure 7a shows the combined loss of 2DMA-Net, the cycle-consistency loss, and the full loss of CycleGAN during the training process. It can be seen that the losses decrease gradually and remain unchanged from about 200 epochs, demonstrating the favorable convergence performance of 2DMA-Net and CycleGAN. Figure 7b shows the discrimination probability curves of the discriminator $D_Z$ on the true spectrum and the fake spectrum. The discrimination probability of 1 indicates that the discrimination results are true and the discrimination probability of 0 indicates that the discrimination results are fake. It can be seen that the discrimination probability of $D_Z$ simultaneously increases to 1 on the true spectrum and decreases to 0 on the fake spectrum. The increasing capacity of the discriminator $D_Z$ to distinguish between the true and fake spectrums indicates the increasing capacity of the generator $G_{YZ}$ to map the low-accuracy clutter spectrum to the high-accuracy clutter spectrum, hence increasing the following CNCM estimation accuracy.

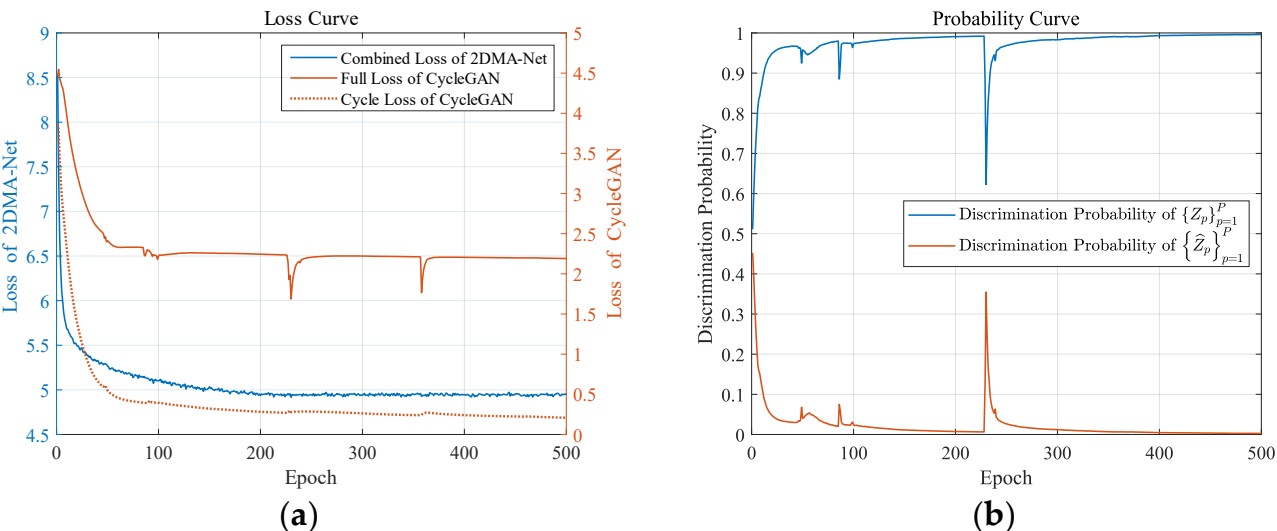

**Figure 7.** Network training convergence results: (**a**) training loss curves of 2DMA-Net and CycleGAN and (**b**) discrimination probability curves of the discriminator $G_Z$.

*4.2. Clutter Spectrum Estimation*

In this subsection, the clutter spectrum estimation results of the proposed DU-CG network are presented under different situations. For comparison, the results obtained via MMV-FOCUSS, MMV-FCSBL, and MMV-ADMM are also shown. As a reference, the MVDR clutter spectrum is calculated based on the theoretical CNCM.

First, Figure 8 shows the estimation results using different methods in the ideal case, i.e., the case with the clutter ridge slope as 1, non-side-looking angle as 0, and no ICM or element amplitude/phase error. It can be seen that as the clutter has a high sparsity in the ideal case, these methods can all estimate the clutter spectrum accurately. As a module of DU-CG, the results obtained using 2DMA-Net have relatively low accuracy where the clutter ridge is broadened. However, as the clutter feature is clearly achieved, based on the

output of 2DMA-Net, the CycleGAN in DU-CG can successfully obtain a high-accuracy clutter spectrum estimation.

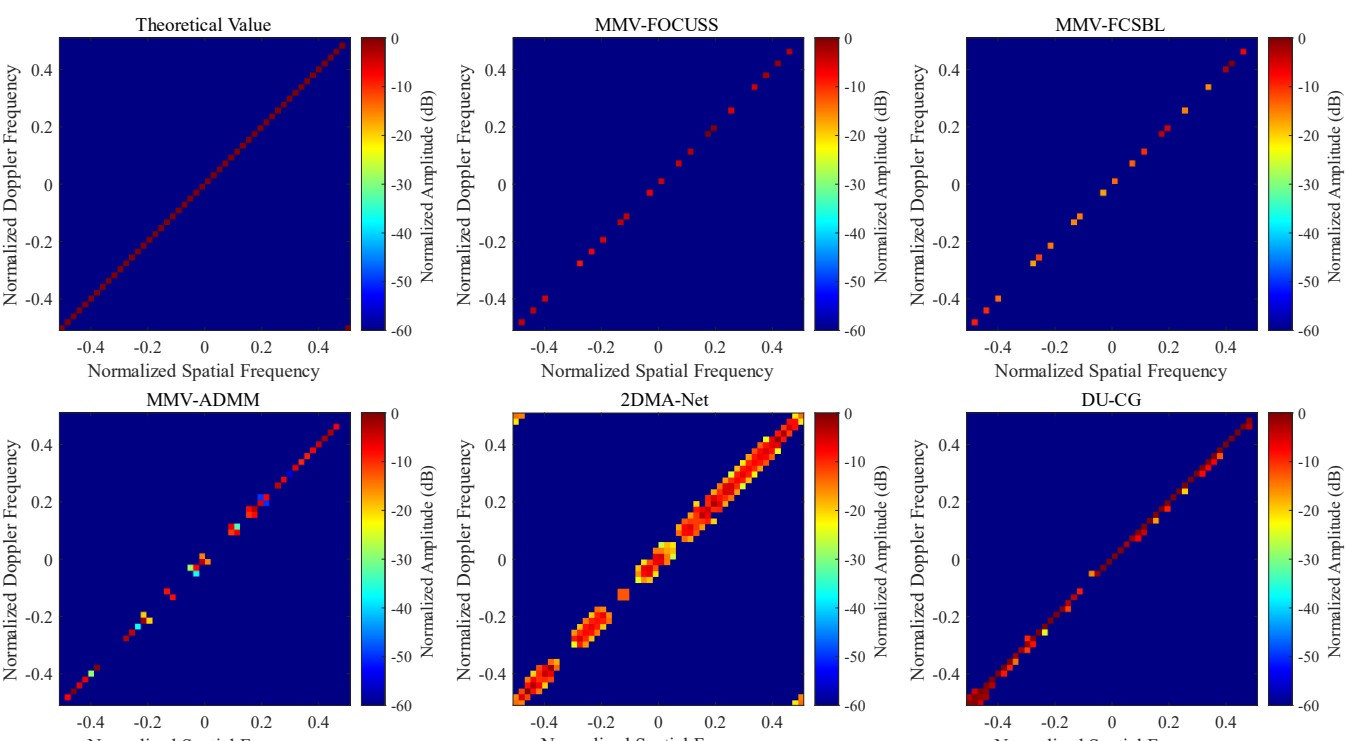

**Figure 8.** Clutter spectrum estimation results obtained using different methods in the ideal case.

Then, Figures 9 and 10 show the estimation results in the non-ideal cases, where the clutter ridge slope is changed to 1.34 and the non-side-looking angle is changed to 16.5°, respectively. It can be seen that, as the clutter sparsity deteriorates in these two cases, the estimation accuracy of typical MMV-SR algorithms degrades significantly. The clutter ridges obtained by these algorithms are broadened and some significant interferences deviating from the clutter ridges are generated. 2DMA-Net can obtain the low-accuracy clutter spectrum estimation with clear clutter features, and thus, based on the output of 2DMA-Net, a high-accuracy clutter spectrum estimation can be obtained by CycleGAN, which is consistent with the reference. These results demonstrate that in the non-ideal cases, typical MMV-SR algorithms are seriously affected by the deteriorated clutter sparsity, whereas the proposed model-driven and data-driven DU-CG network can effectively overcome this problem and adaptively extract the clutter feature to obtain the high-accuracy clutter spectrum estimation.

Furthermore, Figures 11 and 12 show the estimation results of DU-CG in the other two non-ideal cases, where the ICM is set as 0.5 m/s and the element amplitude/phase error is set as 0.14/3.83°. It can be seen that in the presence of ICM, the clutter is broadened along the Doppler dimension due to the temporal decorrelation problem, leading to the damaged clutter sparsity. Hence, typical MMV-SR algorithms will have decreased clutter spectrum estimation accuracy. In the presence of an array element amplitude/phase error, as the SR estimation model and the clutter sparsity are both damaged, the performance of typical MMV-SR algorithms degrades significantly. However, although the performance of 2DMA-Net also degrades in these two cases, the clutter feature is maintained. Then, as CycleGAN can reduce the width of the clutter ridge and suppress the discrete interferences in the space–time domain, a high-accuracy clutter spectrum closest to the reference can still be obtained by the DU-CG network.

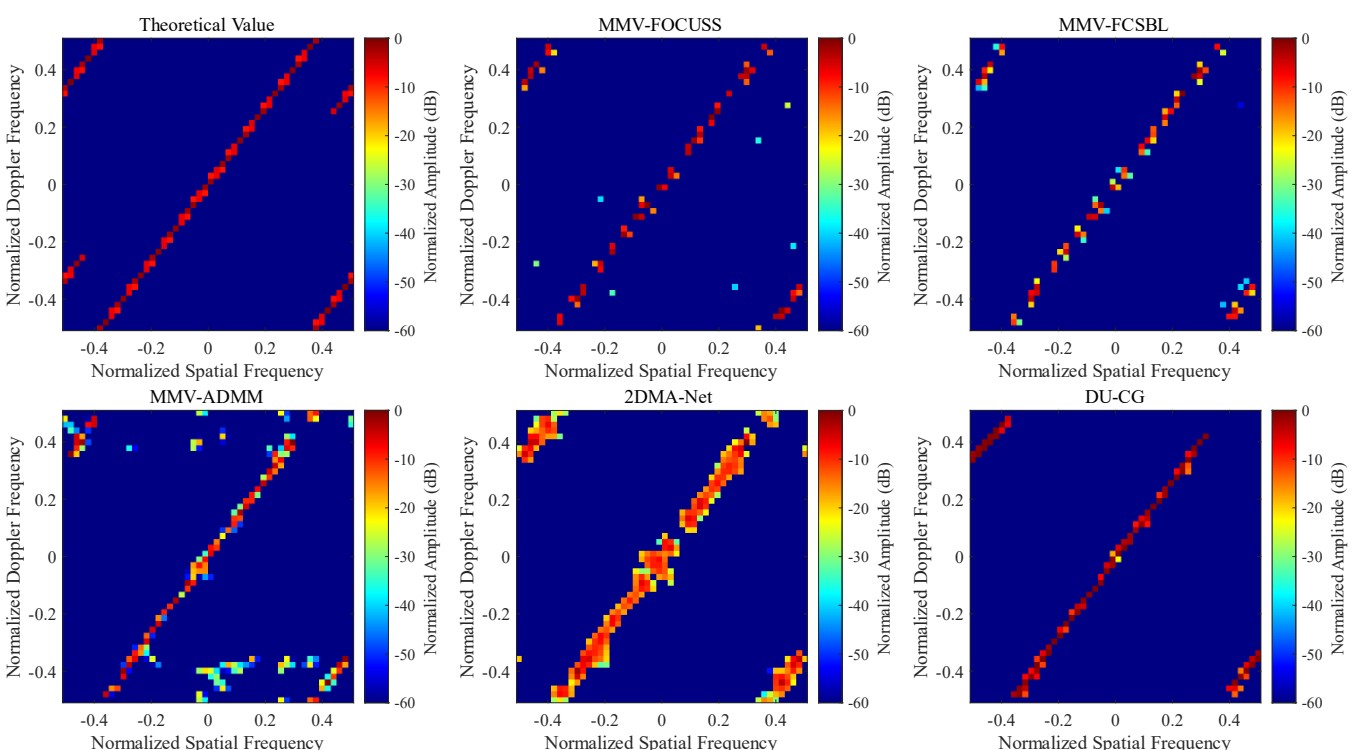

**Figure 9.** Clutter spectrum estimation results obtained using different methods in the non-ideal case with the clutter ridge slope as 1.34.

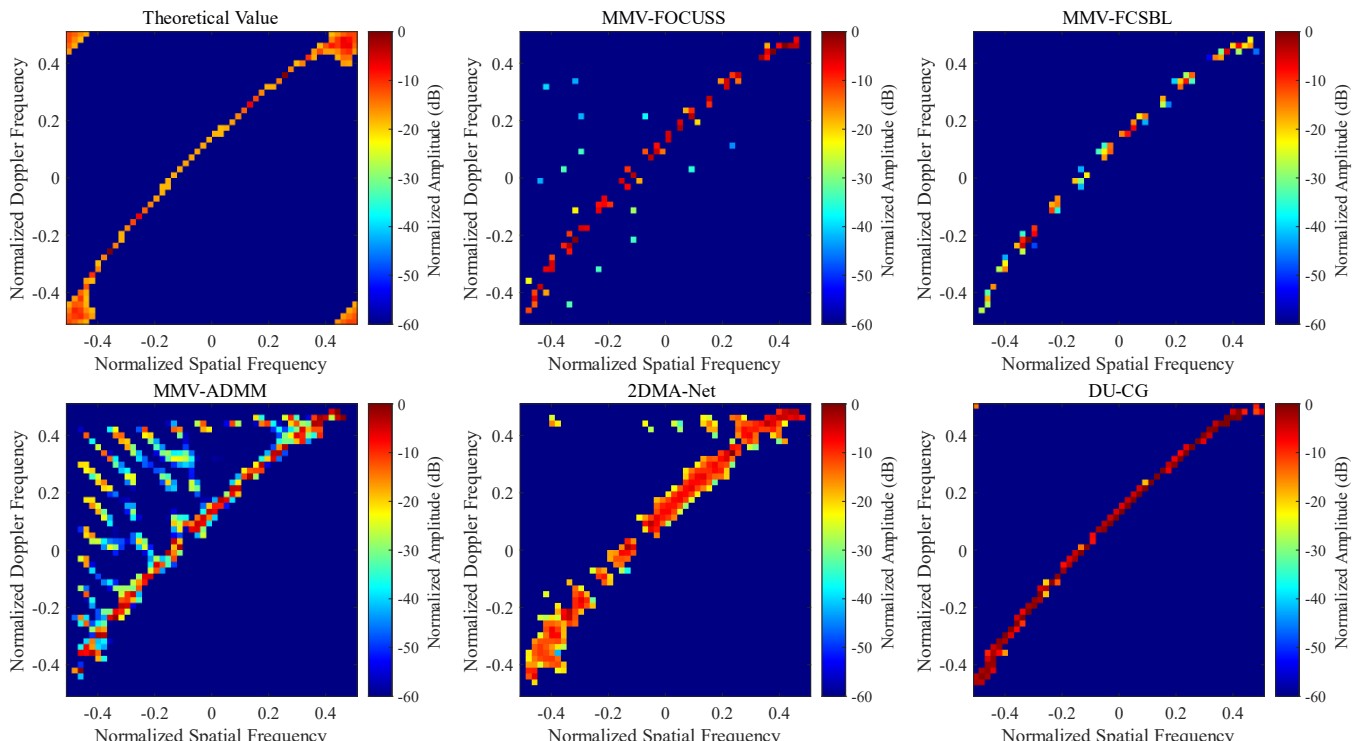

**Figure 10.** Clutter spectrum estimation results obtained using different methods in the non-ideal case with a non-side-looking angle of 16.5°.

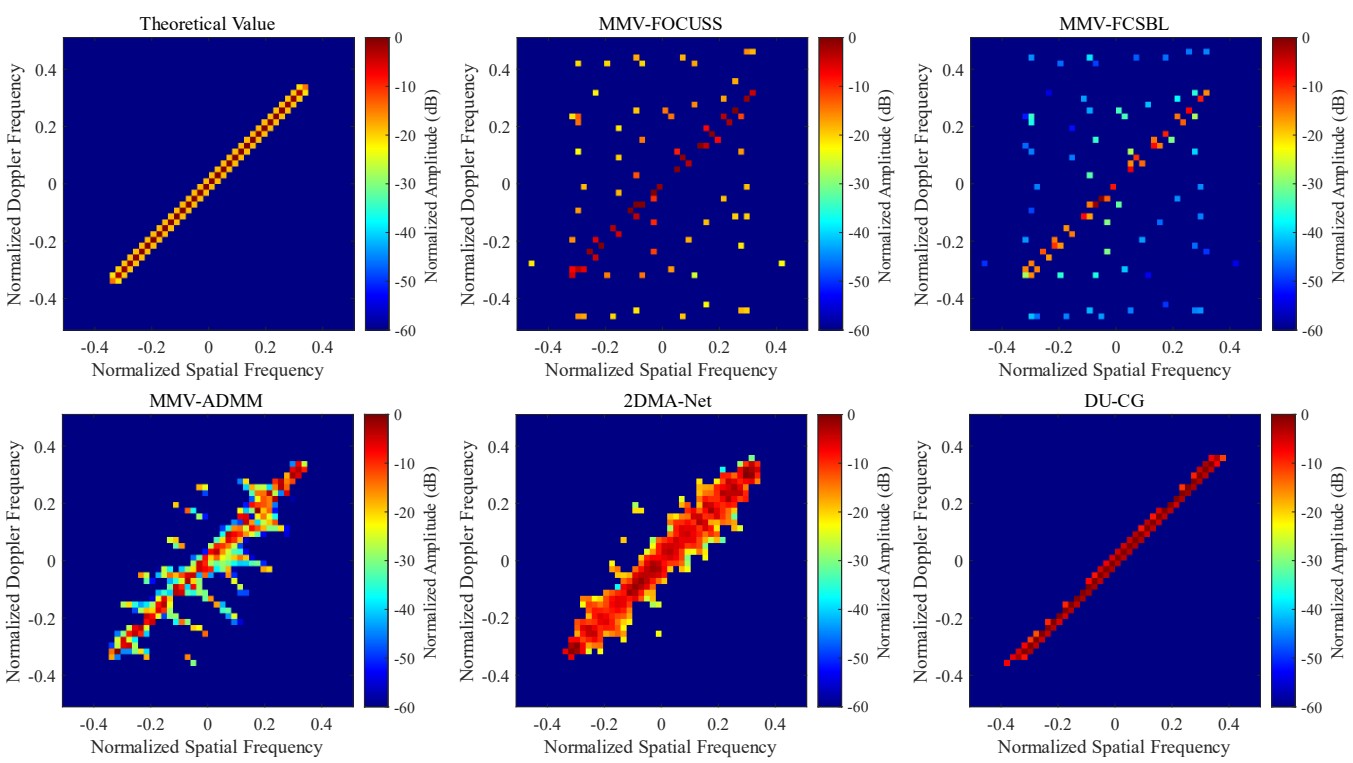

**Figure 11.** Clutter spectrum estimation results obtained using different methods in the non-ideal case with the ICM as 0.5 m/s.

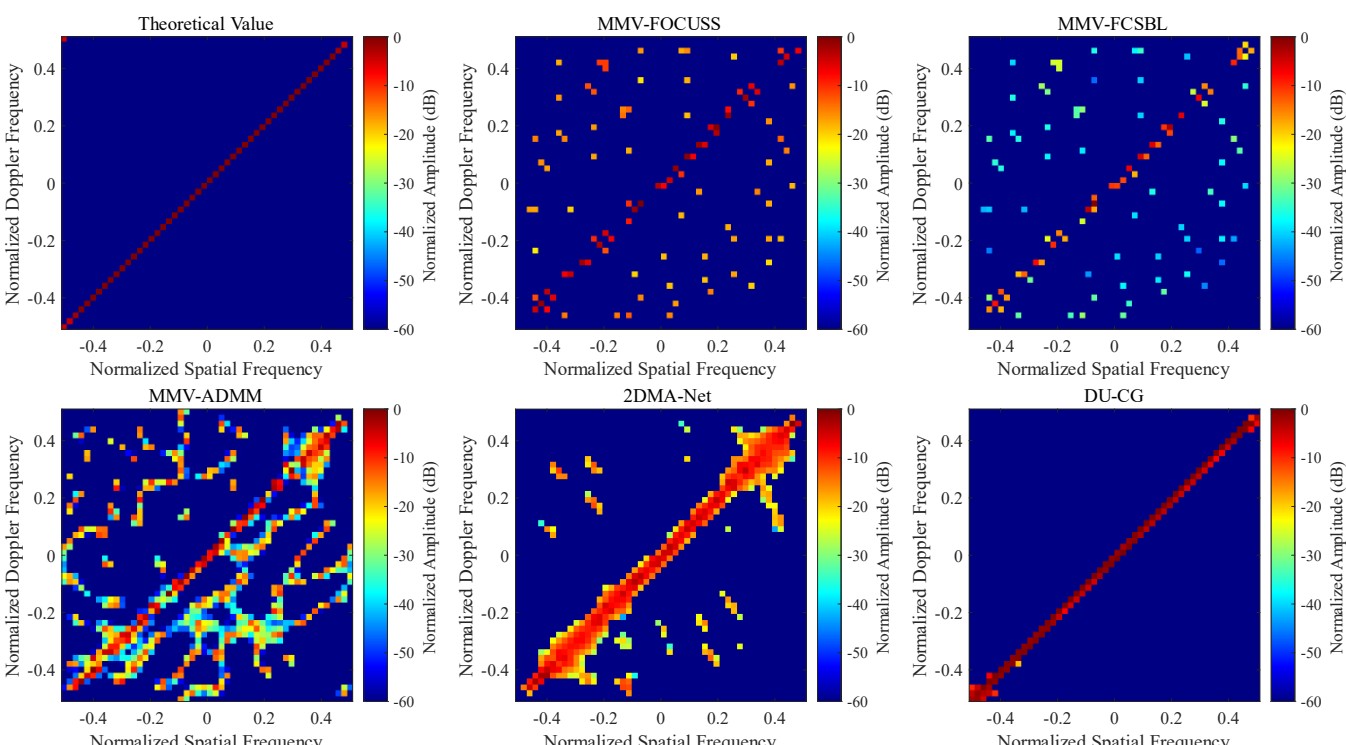

**Figure 12.** Clutter spectrum estimation results obtained using different methods in the non-ideal case with the element amplitude/phase error as 0.14/3.83°.

Finally, all the above-mentioned non-ideal factors are considered, giving the results shown in Figure 13, where the clutter ridge slope is 0.67, the non-side-looking angle is 15.50°, the ICM is 0.24 m/s, and the array element amplitude/phase error is 0.10/4°. The results show that in such a complicated case, the performance of typical MMV-SR

algorithms degrades significantly, the clutter ridge feature distorts severely, and a lot of false peaks appear in the space–time domain. As the proposed DU-CG network can adaptively acquire the clutter features and filter out the interferences caused by non-ideal factors, an accurate estimation of the clutter spectrum is still achieved.

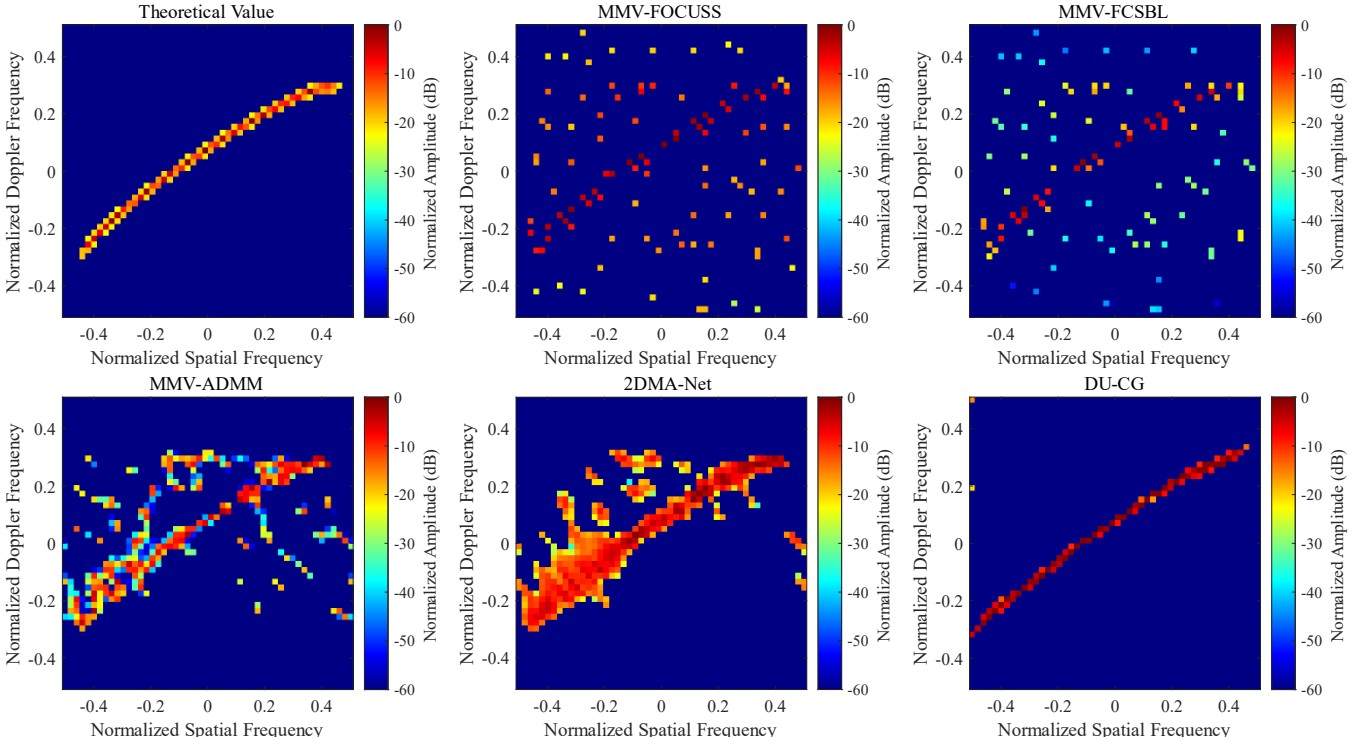

**Figure 13.** Clutter spectrum estimation results obtained using different methods in the non-ideal case with the clutter ridge slope as 0.67, the non-side-looking angle as 15.50°, the ICM as 0.24 m/s, and the array element amplitude/phase error as 0.10/4°.

### 4.3. Clutter Suppression Performance

In this subsection, the clutter suppression performance of different STAP methods is compared using the SCNR loss as the indicator. Keeping the spatial frequency of the target as 0 and linearly varying its normalized Doppler frequency in the range $[-0.5, 0.5]$, the obtained results are shown in Figure 14, where the subfigures (a)–(f), respectively, correspond to Figures 8–13.

The comparison in Figure 14a shows that, in the ideal case, the MMV-FOCUSS-STAP method and MMV-FCSBL-STAP method can achieve the best clutter suppression performance, whereas the proposed DU-CG-STAP method can obtain slightly worse suboptimal performance, which is better than the MMV-ADMM-STAP method. The comparisons in Figure 14b,c show that the clutter suppression performance of typical MMV-SR-STAP methods degrades with the clutter sparsity deterioration, which is manifested by the broadened notch in the zero-Doppler region and the false notches deviating from the clutter ridge. The proposed DU-CG-STAP method can obtain a narrower clutter suppression notch and avoid false notches. The comparisons in Figure 14d,e show that in the presence of ICM and array element amplitude/phase error, typical MMV-SR-STAP methods have significant SCNR losses in almost the entire Doppler frequency range, hence they will suppress not only the clutter but also the target, resulting in low target-detection performance. The proposed DU-CG-STAP method can form an effective suppression notch for the clutter and maintain the power for the target, hence it has a higher performance. The comparison in Figure 14f shows that under conditions with all considered non-ideal factors, compared to typical MMV-SR-STAP methods, the proposed DU-CG-STAP method can still obtain a high clutter suppression performance that is close to the theoretical optimal STAP.

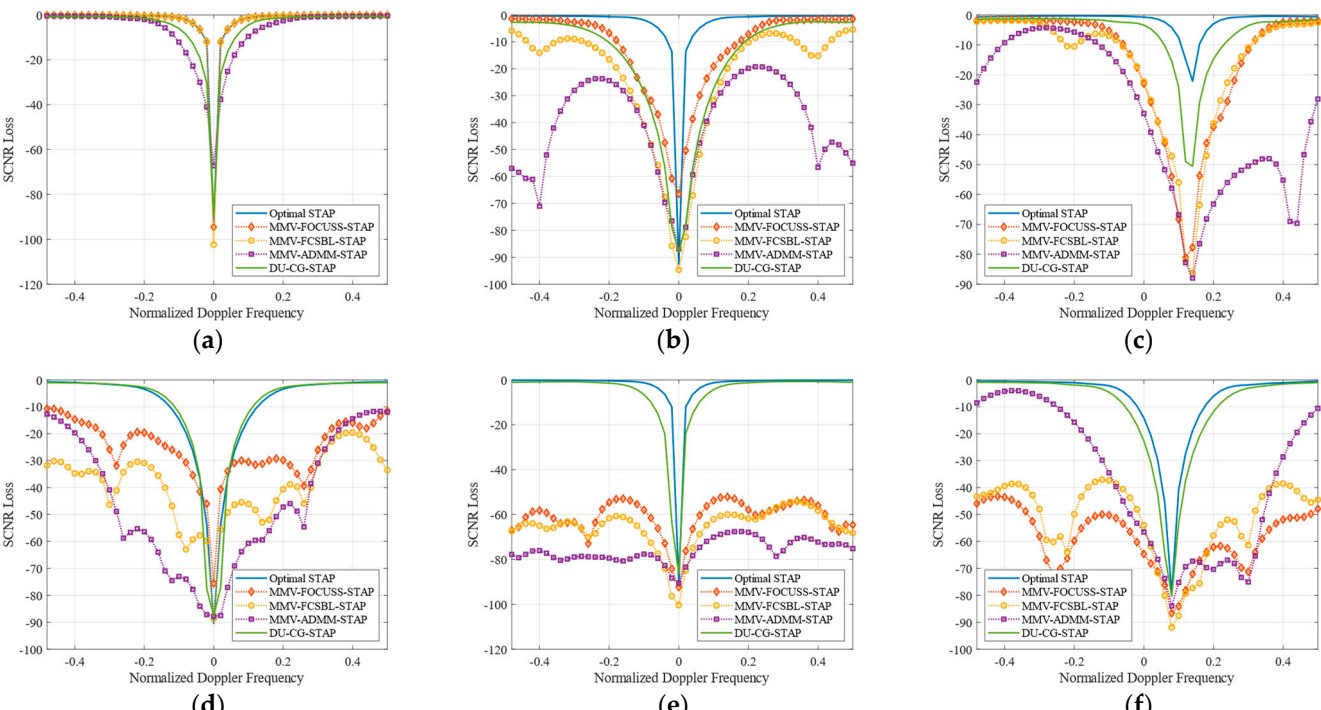

**Figure 14.** SCNR loss curves of different methods for a target with zero spatial frequency and varying Doppler frequency; subfigures (**a**–**f**) correspond to Figures 8–13, respectively.

### 4.4. Computational Complexity Analysis

In this subsection, the computational complexity of the DU-CG network is analyzed and compared with MMV-FOCUSS, MMV-FCSBL, and MMV-ADMM. It should be noted that when applied, the computational complexity of DU-CG is the sum of those of 2DMA-Net and the generator G$_{YZ}$ in CycleGAN. With the only difference in the iterative parameters, the computations of 2DMA-Net and the 2D-MMV-ADMM algorithm are the same. Thus, with the same number of network layers and iterations, 2DMA-Net and the 2D-MMV-ADMM algorithm will have the same computational complexity. Using the multiplication numbers as the indictor, the computational complexities of different algorithms are given in Table 2.

**Table 2.** Computational complexity of different algorithms.

| Algorithm | Computational Complexities |
|---|---|
| MMV-FOCUSS | $O\left(\left(NMN_dN_sL + (NM)^3 + 2(NM)^2N_dN_s + NM(N_dN_s)^2\right)K\right)$ |
| MMV-FCSBL | $O\left(\left(5N_dN_s(NM)^2 + (NM)^3 + (2N_dN_sL + 4N_dN_s + L)(NM) + 3N_dN_s + L\right)K\right)$ |
| MMV-ADMM | $O\left(\left(2NMN_dN_sL + (N_dN_s)^2L + 3NML + N_dN_sL\right)K\right)$ |
| 2DMA-Net | $O\left((2N_dN_sL + NM(N_d + N_s)L + (N + M)N_dN_sL)K\right)$ |
| Generator G$_{YX}$ | $O\left(\sum_{e=1}^{E} c_ef_e^2n_eN_dN_s\right) = O(27516N_dN_s)$ |

According to Table 2, the computational complexities of different algorithms under different conditions are shown in Figure 15. Figure 15a corresponds to the conditions of $N = N_d/5 = 10$, $M = N_s/5 = 10$, and the number of algorithm iterations or network layers $K$ varying from 20 to 50 with a step of 5. Figure 15b corresponds to the conditions of $M = N = N_d/5 = N_s/5$ varying from 5 to 30 with a step of 5 and the number of algorithm iterations or network layers $K$ of MMV-FOCUSS, MMV-FCSBL, MMV-ADMM, and the DU-CG network as 200, 30, 2000, and 30 (which are determined considering their convergence performance). The comparisons show that the computational complexities of 2DMA-Net and the generator G$_{YX}$ in CycleGAN are much lower than the other methods.

Hence, the proposed DU-CG network can always obtain a faster convergence speed under different conditions.

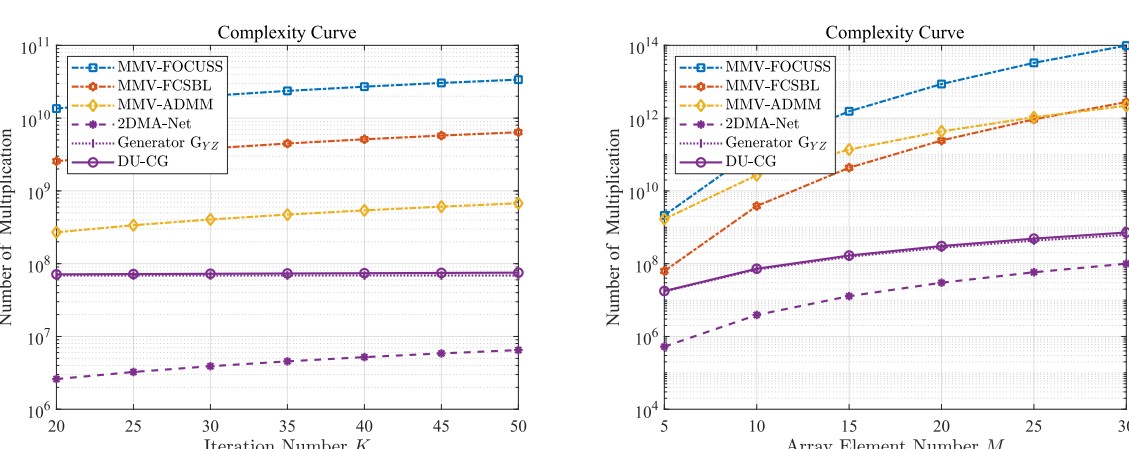

**Figure 15.** Computational complexities of different algorithms under different conditions: (**a**) the computational complexity varies with the iteration number and (**b**) the computational complexity varies with the array element number.

### 4.5. Rationality of DU-CG-STAP

In the CycleGAN of DU-CG-STAP, the discriminator $D_Z$ learns the features of the high-accuracy clutter spectrum with the capacity to discriminate the true and fake spectra, which is continuously improved based on the unpaired dataset. At the same time, the generator $G_{YZ}$ is committed to mapping the low-accuracy clutter spectrum into the high-accuracy domain. If the low-accuracy clutter spectrum is provided with poor quality, it will be difficult for CycleGAN to extract the clutter features and complete the high-accuracy reconstruction task in the unsupervised training process. Hence, to illustrate the rationality of the DU-CG-STAP processing framework, the following results are provided.

In the proposed processing framework, the low-accuracy clutter spectrum dataset is generated by the self-supervised trained 2DMA-Net and used as the input data for Cycle-GAN. If the low-accuracy clutter spectrum dataset is generated by some low-accuracy and low-resolution methods, the high-resolution clutter spectrum reconstruction performance of CycleGAN will seriously degrade. For example, with the low-accuracy clutter spectrum obtained using the Fourier transform and the MVDR methods that were conducted on the raw radar data, the low-accuracy and high-accuracy mapping results of CycleGAN under different conditions are obtained and shown in Figure 16a,b, where the same network scale and training process with the proposed method are used.

It can be seen in Figure 16a that due to the high sidelobes in the Fourier clutter spectrum, the generator $G_{YZ}$ of CycleGAN incorrectly extracts many high values, resulting in significant distortions of the clutter features. It can be seen in Figure 16b that with a small amount of training range cells, the clutter ridge obtained by the MVDR algorithm broadens and some noises exist in the clutter spectrum. Hence, even though the generator $G_{YZ}$ can extract the clutter features, it cannot effectively reduce the clutter ridge width and suppress the noisy spectrum component.

On the contrary, by generating the low-accuracy clutter spectrum dataset via the MMV-ADMM algorithm and the trained 2DMA-Net, the low-accuracy and high-accuracy mapping results of CycleGAN are obtained and shown in Figure 16c,d. Since the clutter spectra obtained by these two approaches have no obvious sidelobe/noises and the features of the clutter ridge are clear, CycleGAN can filter out the interferences caused by non-ideal factors in the unsupervised training process so as to complete the high-accuracy clutter spectrum reconstruction task.

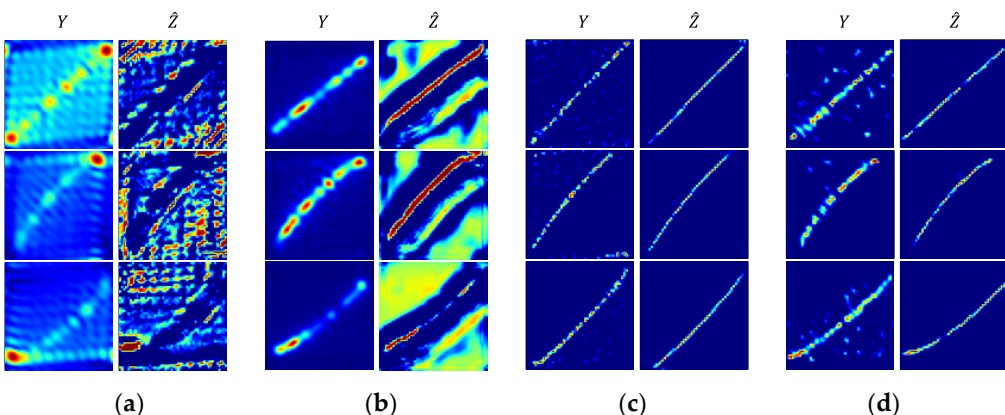

**Figure 16.** Low-accuracy and high-accuracy mapping results of CycleGAN with the low-accuracy clutter spectrum dataset obtained by (**a**) Fourier transform, (**b**) MVDR algorithm, (**c**) MMV-ADMM algorithm, and (**d**) the trained 2DMA-Net.

## 5. Conclusions

To solve the problems of high computational complexity, parameter-setting difficulties, and degraded performance caused by non-ideal factors in the conventional SR-STAP methods for airborne radar moving-target detection, a novel DU-CG-STAP method has been proposed in this paper. The processing framework, network structure, dataset construction, and training methods of the proposed method have been introduced in detail. The simulation results obtained under different situations have shown that compared to existing typical SR-STAP methods, the proposed method can simultaneously improve the clutter spectrum estimation accuracy and reduce the computational complexity, thus achieving a higher clutter suppression and target detection performance. In future work, we will focus on the improvement of the SR-based DU-Nets and the image-enhancement DNNs for STAP applications.

**Author Contributions:** Conceptualization, B.Z., W.F. and X.W.; methodology, B.Z., X.W. and W.F.; software, B.Z., H.Z. and F.L.; validation, B.Z., H.Z. and F.L.; formal analysis, W.F. and X.W.; investigation, W.F. and X.W.; resources, W.F. and H.Z.; data curation, F.L. and B.Z.; writing—original draft preparation, B.Z.; writing—review and editing, W.F. and X.W.; visualization, B.Z., H.Z. and F.L.; supervision, W.F. and H.Z.; project administration, W.F.; funding acquisition, W.F. All authors have read and agreed to the published version of the manuscript.

**Funding:** This work was supported by National Natural Science Foundation of China, No. 62001507, and the Young Talent fund of University Association for Science and Technology in Shaanxi, China, No. 20210106.

**Conflicts of Interest:** The authors declare no conflict of interest.

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
