# Peer review of "DU-CG-STAP Method Based on Sparse Recovery and Unsupervised Learning for Airborne Radar Clutter Suppression"

_remotesensing, doi:10.3390/rs14143472_

Round 1
Reviewer 1 Report
This paper describes a novel framework to suppress clutter given the geometry of a ULA from an airborne platform. The paper is well written and does a good job covering a variety of practical simulated cases, comparing the proposed method with other state of the art methods.
Sufficient detail is provided to describe the algorithm and overall framework.
The authors do a nice job considering performance and calculation cost.
It would be nice to have some real data but that obviously can be expensive for this type of work, so in my mind it is acceptable to have simulation only data for this paper.
Perhaps I missed it but it seems like it would be important to have a good selection of your sparse clutter samples. How do the authors propose selecting these sparse clutter samples that would not contain a desired moving target? How does the algorithm perform if the clutter samples are contaminated by moving target? Maybe the authors could address that in the text if they have not already.
The title of the paper is "for Airborne Radar Moving Target Detection" and while the framework does allow for subsequent moving target detection the bulk of the work is dedicated to clutter suppression. I think it would be better to have the title of the paper talk say "for Airborne Radar Clutter Suppression". If the authors would like the title to remain as is, perhaps they can rework the analysis and conclusions to show how the better clutter suppression enables superior moving target detection.
Overall a very nice, well written, paper.
Reviewer 2 Report
The paper presented is well written and has the high scientific soundness, but there are some questions and recommendations to the the authors.
1) What does it mean: "... combination of the spatial and temporal two-dimensional information"? The authors wrote in Lines 184-185: "...discretizing the entire space-timel domain...".
2) How much is parameter ro in (18)?
3) How canYyou characterize the parameter betha in system (20)? How is this system solved? Is it a bad-conditioned one or not?
4) What happens if the simulation parameters in Table 1 increaese dramatically? Please , could You briefly dscribe the algorithm of selecting those values.
